# LEARNING EQUIVARIANT TENSOR FUNCTIONS WITH APPLICATIONS TO SPARSE VECTOR RECOVERY

## ABSTRACT

This work characterizes equivariant polynomial functions from tuples of tensor inputs to tensor outputs. Loosely motivated by physics, we focus on equivariant functions with respect to the diagonal action of the orthogonal group on tensors. We show how to extend this characterization to other linear algebraic groups, including the Lorentz and symplectic groups.

Our goal behind these characterizations is to define equivariant machine learning models. In particular, we focus on the sparse vector estimation problem. This problem has been broadly studied in the theoretical computer science literature, and explicit spectral methods, derived by techniques from sum-of-squares, can be shown to recover sparse vectors under certain assumptions. Our numerical results show that the proposed equivariant machine learning models can learn spectral methods that outperform the best theoretically known spectral methods in some regimes. The experiments also suggest that learned spectral methods can solve the problem in settings that have yet to be theoretically analyzed.

This is an example of a promising direction in which theory can inform machine learning models and machine learning models can inform theory.

## 1 INTRODUCTION

Many recent theoretical and applied efforts have focused on the implementation of symmetries and other structural constraints in the design of machine learning models. This is the case of graph neural networks Scarselli et al. (2008); Maron et al. (2019), geometric deep learning Bronstein et al. (2021); Weiler et al. (2021), and AI for science Zhang et al. (2023). The goal is to design a hypothesis class of functions with good *inductive bias* that is aligned with the theoretical framework of the physical, mathematical, or algorithmic objects it aims to represent. This includes respecting coordinate freedoms Villar et al. (2023a), conservation laws Alet et al. (2021), or internal symmetries (e.g. in the implicit neural representations framework Lim et al. (2023)). Symmetries have also been used to provide interpretability to learned data representations Suau et al. (2023); Gupta et al. (2023). Mathematically, it has been shown that imposing symmetries can improve the generalization error and sample complexity of machine learning models Elesedy (2021b); Wang et al. (2021b); Elesedy (2021a); Bietti et al. (2021); Petrache & Trivedi (2024); Tahmasebi & Jegelka (2023); Huang et al. (2024).

In this work, we focus on implementing equivariant functions of tensor inputs and tensor outputs. Tensors are the subjects of study in many theoretical computer science and applied mathematics problems, including tensor factorization or decomposition Rabanser et al. (2017), and planted tensor models Hopkins et al. (2016). Many of the algorithms to address these problems have underlying symmetries like the ones we study here. Tensors also have a broad set of uses in the natural sciences, where tensor-valued data appears as polarizations Melrose & Stoneham (1977), permeabilities Durlofsky (1991), and stresses Levitas et al. (2019), for instance. Finally, tensors are the preferred way to represent data for machine learning, where both models and multidimensional data with batches, channels, etc, are implemented as tensors. Sometimes, these tensor objects have underlying symmetries, like in implicit neural representations Li et al. (2022).

In this paper, we consider classical Lie groups acting diagonally on tensors. The groups we study arise naturally in physics and other settings, including the orthogonal group $O(d)$ (which typically appears in coordinate transformations), the indefinite orthogonal group $O(s, k - s)$ (which includes

as a particular case the Lorentz group, a fundamental group for special relativity), and the symplectic group $Sp(d)$ (the underlying group in much of classical and quantum mechanics).

A variety of methods can be used for implementing invariances or equivariances, including group convolutions Cohen & Welling (2016; 2017); Wang et al. (2021a), irreducible representations Fuchs et al. (2020); Kondor (2018); Weiler et al. (2018); Cohen et al. (2018); Weiler & Cesa (2019), constraints on optimization Finzi et al. (2021), canonicalization Kaba et al. (2023), and invariant theory Gripaios et al. (2021); Haddadin (2022); Villar et al. (2021); Blum-Smith & Villar (2023); Villar et al. (2023b). This work is closer to the line of research that constructs explicit equivariant functions from invariant features, and it generalizes results from Villar et al. (2021) to tensors.

Closest to us is the concurrent work of Kunisky, Moore, and Wein on tensor cumulants Kunisky et al. (2024). Their focus is primarily on symmetric tensors, although they show that $O(d)$-invariant polynomials on symmetric tensors can be turned into $O(d)$-invariant polynomials over general tensors by symmetrizing over $O(d)$. Our results for equivariant tensor polynomials are slightly more general for inputs of tensors of different orders and parities as well as handling the indefinite orthogonal group and the symplectic group.

As an application, we consider the problem of sparse vector estimation. Let there be a sparse vector $v_0$ and vectors $v_1, \ldots, v_{d-1}$ sampled independently at random from some prior distribution. If we are given an orthonormal basis $w_0, \ldots, w_{d-1}$ of span$(\{v_0, \cdots, v_{d-1}\})$, can we recover the sparse vector $v_0$? The problem has roots in tensor PCA, also known as the spiked tensor model Montanari & Richard (2014), as well as dictionary learning Spielman et al. (2012). Solutions for this problem using sum-of-squares and spectral methods were investigated in Barak et al. (2014) and further improvements were made in Hopkins et al. (2015; 2016); Ge & Ma (2015); Mao & Wein (2022).

In Hopkins et al. (2016), the authors propose an algorithm that constructs a $d \times d$ matrix $A$, then uses the top eigenvector of $A$ to estimate $v_0$. They prove that this method recovers the planted vector under certain assumptions. A crucial observation is that the input to the problem is *any* orthonormal basis $w_0, \ldots, w_{d-1}$ of span$(\{v_0, \ldots, v_{d-1}\})$. This implies that a function that estimates $v_0$ should be invariant to certain transformations of the input basis. In this paper, we learn algorithms to recover planted sparse vectors from data. To this end, we use a machine learning model that learns an equivariant 2-tensor $\hat{A}$ from data, and the estimator for the planted sparse vector $\hat{v}$ is obtained using $\hat{A}$'s top eigenvector following the same procedure as Hopkins et al. (2016). We empirically show that this approach can learn spectral methods for sparse vector recovery that can work under more general assumptions than the state-of-the-art.

The approach employed here can be seen as a particular case of the more ambitious research program of algorithmic alignment or algorithmic reasoning Veličković & Blundell (2021); Gavranović et al. (2024). In algorithmic alignment contexts, researchers design machine learning models that match known algorithmic strategies that are known to succeed at solving certain problems, such as dynamic programming Xu et al. (2020); Dudzik & Veličković (2022). Here, the machine learning approach is structurally aligned with equivariant spectral methods arising from sum-of-squares. We believe that the connection between sum-of-squares and machine learning (in particular equivariant machine learning and graph neural networks) is a promising direction to be explored.

## 1.1 OUR CONTRIBUTIONS

We provide a generic recipe to define equivariant machine learning models mapping from tensors to tensors. To this end, we give explicit parameterizations for polynomials (Sec. 3) and analytic functions with globally convergent Taylor series (Sec. 4) from tuples of tensor inputs to tensor outputs that are equivariant with respect to the orthogonal (Sec. 3), indefinite orthogonal (which includes Lorentz), and symplectic groups (Sec. 4). This generalizes the existing results of Villar et al. (2021) and leverages the tensor equivariant theory Appleby et al. (1987); Jeffreys (1973); Roe Goodman (2009) into a format useful for machine learning frameworks. On first reading and for those primarily interested in practical applications, we suggest focusing on Corollary 1 in Section 3 from which all our experiments follow.

As a proof of concept, we applied the resulting models to learning algorithms for the sparse vector recovery problem (Section 5). Our learned algorithm outperforms state-of-the-art sum-of-squares methods for this problem in several regimes with both synthetic data and an image denoising problem.

## 2 DEFINITIONS

To simplify the exposition, we start by focusing on the case of the orthogonal group before extending the result to the indefinite orthogonal and symplectic groups. We consider the orthogonal group $O(d)$, the isometries of Euclidean space $\mathbb{R}^d$ that fix the origin. It acts on vectors and pseudovectors $v \in \mathbb{R}^d$ in the following way:

$$g \cdot v = \det(M(g))^{\frac{1-p}{2}} M(g) v, \tag{1}$$

where $g \in O(d)$, $M(g) \in \mathbb{R}^{d \times d}$ is the standard matrix representation of $g$ (i.e. $M(g)^\top M(g) = \mathbb{I}_d$, where $\mathbb{I}_d$ is the identity matrix), and $p \in \{-1, +1\}$ is the parity of $v$. If $p = +1$ we obtain the standard $O(d)$ action on $\mathbb{R}^d$ *vectors*. If $p = -1$ we obtain the $O(d)$ action on what in physics are known as *pseudovectors*. For a common pseudovector, consider a rotating Ferris wheel with angular velocity whose direction is given by the right-hand rule. A reflection of the wheel, which will have $\det(M(g)) = -1$ in (1), does not change the direction of rotation or, therefore, the direction of the angular velocity.

**Definition 1** ($k_{(p)}$-tensors)**.** We define the space of $1_{(p)}$-*tensors* to be $\mathbb{R}^d$ equipped with the action $O(d)$ defined by (1). If $v_i$ is a $1_{(p_i)}$-tensor for $i = 1, \ldots, k$, then $a := v_1 \otimes \ldots \otimes v_k \in (\mathbb{R}^d)^{\otimes k}$ is a *rank-1 $k_{(p)}$-tensor*, where $p = \prod_{i=1}^k p_i$ and the action of $O(d)$ is the diagonal action:

$$g \cdot (v_1 \otimes \ldots \otimes v_k) = (g \cdot v_1) \otimes \ldots \otimes (g \cdot v_k). \tag{2}$$

This definition generalizes to higher rank $k_{(p)}$-tensors by linearity (see (5) below). The space of $k_{(p)}$-tensors in $d$ dimensions is denoted $\mathcal{T}_k(\mathbb{R}^d, p)$.

**Remark 1.** Note that the definition of $k_{(p)}$-tensor includes the selection of the $O(d)$-action on the tensor. In this way, if $a$ is a $k_{(+)}$-tensor, we can see $a$ as a $k_{(-)}$-tensor by redefining the action as $g \cdot a = \det(M(g))(g \cdot a)$, where on the left we have the action as a $k_{(-)}$-tensor and on the right the action as a $k_{(+)}$-tensor.

**Definition 2** (Einstein summation notation)**.** Suppose that $a$ is a $k_{(p)}$-tensor. Let $[a]_{i_1, \ldots, i_k}$ denote the $(i_1, \ldots, i_k)$-th entry of $a$, where $i_1, \ldots, i_k$ range from 1 to $d$. The *Einstein summation notation* is used to represent tensor products[1] where repeated indices are summed over. In each product, a given index can appear either exactly once, in which case it appears in the result, or exactly twice, in which case it is summed over and does not appear in the result.

For example, in Einstein summation notation, the product of two $2_{(+)}$-tensors (i.e., the matrix product $ab$ of two $d \times d$ matrices $a$ and $b$) is written as

$$[a\,b]_{i,j} = [a]_{i,\ell}\,[b]_{\ell,j} := \sum_{\ell=1}^d [a]_{i,\ell}\,[b]_{\ell,j}. \tag{3}$$

Using Einstein summation notation, the action of $g \in O(d)$ on rank-1 tensors can be extended to general tensors by linearity by expressing $b \in \mathcal{T}_k(\mathbb{R}^d, p)$ as a linear combination of (rank-1) standard basis tensors $e_{i_1, \ldots, i_k} = e_{i_1} \otimes \cdots \otimes e_{i_k}$, where $[e_i]_i = 1$ and $[e_i]_j = 0$ for $i \neq j$

$$[g \cdot b]_{i_1, \ldots, i_k} = [b]_{j_1, \ldots, j_k}[g \cdot (e_{j_1} \otimes \cdots \otimes e_{j_k})]_{i_1, \ldots, i_k} = [b]_{j_1, \ldots, j_k}[g \cdot e_{j_1}]_{i_1} \cdots [g \cdot e_{j_k}]_{i_k}. \tag{4}$$

Note that the action (1) on a $k_{(p)}$-tensor $b$ can be written as

$$[g \cdot b]_{i_1, \ldots, i_k} = \det(M(g))^{\frac{1-p}{2}} [b]_{j_1, \ldots, j_k} [M(g)]_{i_1, j_1} \cdots [M(g)]_{i_k, j_k} \tag{5}$$

for all $g \in O(d)$. For example, a $2_{(+)}$-tensor has the transformation property $[g \cdot b]_{i,j} = [b]_{k,\ell} [M(g)]_{i,k} [M(g)]_{j,\ell}$, which, in normal matrix notation, is written as $g \cdot b = M(g)\, b\, M(g)^\top$.

When multiple tensors are combined, and all their indices appear in the result, we refer to that as the tensor product or outer product. When indices are summed over, we refer to that as the contraction or scalar product. We will further focus on a specific case of multiple tensor contractions that we will refer to as a $k$-contraction.

---

[1]We will identify vectors with co-vectors in the usual way and will not distinguish lower vs upper scripts.

**Definition 3** (Outer product of tensors). Given $a \in \mathcal{T}_k(\mathbb{R}^d, p)$ and $b \in \mathcal{T}_{k'}(\mathbb{R}^d, p')$, the *outer product*, denoted $a \otimes b$, is a tensor in $\mathcal{T}_{k+k'}(\mathbb{R}^d, p\,p')$ defined as $[a \otimes b]_{i_1,\ldots,i_{k+k'}} = [a]_{i_1,\ldots,i_k} [b]_{i_{k+1},\ldots,i_{k+k'}}$. We write $a^{\otimes k}$ to denote the outer product of $a$ with itself $k$ times and use the convention for $k = 0$ that $a^{\otimes 0} \otimes b = b$.

**Definition 4** ($k$-contraction). Given a tensor $a \in \mathcal{T}_{2k+k'}(\mathbb{R}^d, p)$, the $k$-*contraction* of $a$, denoted $\iota_k(a)$, is the $k'_{(p)}$-tensor defined as follows using Einstein summation:

$$[\iota_k(a)]_{j_1,\ldots,j_{k'}} := [a]_{i_1,\ldots,i_k,i_1,\ldots,i_k,j_1,\ldots,j_{k'}}. \tag{6}$$

For instance, if $a = u \otimes v \otimes x \otimes y \otimes z \in \mathcal{T}_{4+1}(\mathbb{R}^d, p)$ then $\iota_1(a) = \langle u, x\rangle \langle v, y\rangle z$.

Since $k_{(p)}$-tensors are elements of the vector space $(\mathbb{R}^d)^{\otimes k}$, tensor addition and scalar multiplication are defined in the usual way. The final operation on tensors is the permutation of the indices.

**Definition 5** (Permutations of tensor indices). Given $a \in \mathcal{T}_k(\mathbb{R}^d, p)$ and permutation $\sigma \in S_k$, the *permutation of tensor indices* of $a$ by $\sigma$, denoted $a^\sigma$, is defined by

$$[a^\sigma]_{i_1,\ldots,i_k} := [a]_{i_{\sigma^{-1}(1)},\ldots,i_{\sigma^{-1}(k)}}. \tag{7}$$

**Definition 6** (Invariant and equivariant functions). We say that $f : \mathcal{T}_k(\mathbb{R}^d, p) \to \mathcal{T}_{k'}(\mathbb{R}^d, p')$ is $O(d)$-invariant if

$$f(g \cdot a) = f(a), \quad \text{for all} \quad g \in O(d). \tag{8}$$

We say that $f : \mathcal{T}_k(\mathbb{R}^d, p) \to \mathcal{T}_{k'}(\mathbb{R}^d, p')$ is $O(d)$-equivariant if

$$f(g \cdot a) = g \cdot f(a), \quad \text{for all} \quad g \in O(d). \tag{9}$$

If $f$ were instead a function with multiple inputs, then the same group element $g$ would act on all inputs simultaneously.

**Definition 7** (Isotropic tensors). We say that a tensor $a \in \mathcal{T}_k(\mathbb{R}^d, p)$ is $O(d)$-isotropic if $g \cdot a = a$, for all $g \in O(d)$.

There are two special tensors, the Kronecker delta, and the Levi-Civita symbol. These tensors are $O(d)$-isotropic and, as we will show in Section 3, we can construct all $O(d)$-isotropic tensors using only Kronecker deltas and Levi-Civita symbols.

**Definition 8** (Kronecker delta). The *Kronecker delta*, $\delta$, is the $O(d)$-isotropic $2_{(+)}$-tensor such that $[\delta]_{ij} = 1$ if $i = j$ and $0$ otherwise. When considered as a matrix, it is the identity matrix $\mathbb{I}_d$.

**Definition 9** (Levi-Civita symbol). The *Levi-Civita symbol*, $\epsilon$, in dimension $d \geq 2$ is the $O(d)$-isotropic $d_{(-)}$-tensor such that $[\epsilon]_{i_1,\ldots,i_d} = 0$ if any two of the $i_1,\ldots,i_d$ are equal, $[\epsilon]_{i_1,\ldots,i_d} = +1$ if $i_1,\ldots,i_d$ is an even permutation of $1,\ldots,d$, and $[\epsilon]_{i_1,\ldots,i_d} = -1$ if $i_1,\ldots,i_d$ is an odd permutation of $1,\ldots,d$. For example, when $d = 2$ this is simply the matrix $\begin{bmatrix} 0 & 1 \\ -1 & 0 \end{bmatrix}$.

## 3 $O(d)$-EQUIVARIANT POLYNOMIAL FUNCTIONS

In this section, we characterize $O(d)$-equivariant polynomial functions mapping multiple tensor inputs to tensor outputs. This result generalizes the results of Villar et al. (2021) from $1_{(+)}$-tensors to general $k_{(p)}$-tensors in the case of polynomials for the group $O(d)$. On first reading and for those primarily interested in practical applications, we advise focusing on Example 1 and Corollary 1 below.

Each term in the theorem below should be viewed as combining $r$ of the input tensors with the tensor product, then mapping them to the appropriate output with a linear map. Since a linear map between tensors can always be written as a tensor product followed by a sequence of contractions [Dimitrienko (2013),Theorem 5.1], that is what we do with the new tensor $c_{\ell_1,\ldots,\ell_r}$. However, since the function is also $O(d)$-equivariant, the tensor $c_{\ell_1,\ldots,\ell_r}$ must be $O(d)$-isotropic. The theorem merely says that this polynomial is enough to construct all the tensor equivariant polynomials.

**Theorem 1.** Let $f : \prod_{i=1}^{n} \mathcal{T}_{k_i}(\mathbb{R}^d, p_i) \to \mathcal{T}_{k'}(\mathbb{R}^d, p')$ be an $O(d)$-equivariant polynomial function of degree at most $R$. Then we may write $f$ as follows:

$$f(a_1,\ldots,a_n) = \sum_{r=0}^{R} \sum_{1 \leq \ell_1 \leq \cdots \leq \ell_r \leq n} \iota_{k_{\ell_1,\ldots,\ell_r}}(a_{\ell_1} \otimes \ldots \otimes a_{\ell_r} \otimes c_{\ell_1,\ldots,\ell_r}) \tag{10}$$

where $c_{\ell_1,\ldots,\ell_r}$ is an $O(d)$-isotropic $(k_{\ell_1,\ldots,\ell_r} + k')_{(p_{\ell_1,\ldots,\ell_r}\, p')}$-tensor for $k_{\ell_1,\ldots,\ell_r} = \sum_{q=1}^{r} k_{\ell_q}$ and $p_{\ell_1,\ldots,\ell_r} = \prod_{q=1}^{r} p_{\ell_q}$.

Here is an example of Theorem 1 in action, expressing a given equivariant polynomial in terms of invariant functions and tensors. A longer example appears in Appendix E.

**Example 1.** Let $f : \mathcal{T}_1(\mathbb{R}^d, +) \to \mathcal{T}_2(\mathbb{R}^d, +)$ be an $O(d)$-equivariant polynomial of degree at most 2. By Theorem 1, we can write $f$ in the form

$$f(a) = \iota_0\big(a^{\otimes 0} \otimes c_0\big) + \iota_1\big(a^{\otimes 1} \otimes c_1\big) + \iota_2\big(a^{\otimes 2} \otimes c_2\big), \tag{11}$$

where $c_r$ is an $O(d)$-isotropic $(r+2)_{(+)}$-tensor for $r = 0, 1, 2$. Lemma 1 (proven below) characterizes such isotropic tensors $c_r$: $c_0 = \beta_0 \delta$, $c_1 = 0$ is trivial, and $c_2$ is a linear combination of $(\delta^{\otimes 2})^{\sigma}$ for $\sigma \in G_4 = \{\sigma_1, \sigma_2, \sigma_3\}$ where $\sigma_1 := (1,2,3,4)$, $\sigma_2 = (1,3,2,4)$, $\sigma_3 = (1,3,4,2)$, see Appendix D.

Thus the final term $\iota_2\big(a^{\otimes 2} \otimes c_2\big)$ is

$$\iota_2\big(a^{\otimes 2} \otimes \big(\beta_1(\delta^{\otimes 2})^{\sigma_1} + \beta_2(\delta^{\otimes 2})^{\sigma_2} + \beta_3(\delta^{\otimes 2})^{\sigma_3}\big)\big) = \beta_1\langle a, a\rangle + \beta_2 a \otimes a + \beta_3 a \otimes a, \tag{12}$$

where the terms associated with $\beta_2$ and $\beta_3$ are the same due to the symmetry of $a^{\otimes 2}$. We conclude

$$f(a) = \beta_0 \delta + \beta_1 \langle a, a\rangle \delta + \beta_2 a \otimes a, \tag{13}$$

for some scalars $\beta_0$, $\beta_1$, and $\beta_2$.

While the sums in Lemma 1 are over the full symmetric group, in Example 1 we reduced the number of terms by considering the symmetries of the summands. We develop this in detail in Appendix D.

The proof of Theorem 1 is given in Appendix B. The condition that $c_{\ell_1,\ldots,\ell_r}$ is $O(d)$-isotropic is quite restrictive; the following lemma says that all such tensors can be constructed from the Kronecker delta $\delta$ (def. 8) and the Levi-Civita symbol (def. 9). This lemma, originally from Pastori, follows from Jeffreys (1973).

**Lemma 1** (Characterization of $O(d)$-isotropic $k_{(p)}$-tensors). Suppose $c \in \mathcal{T}_k(\mathbb{R}^d, p)$ is $O(d)$-isotropic. Then the following holds:

*Case $p = +1$:* Assume $p = +1$. If $k$ is even, then $c$ can be written in the form

$$c = \sum_{\sigma \in S_k} \alpha_\sigma \Big(\delta^{\otimes \frac{k}{2}}\Big)^{\sigma}, \quad \text{for any } \alpha_\sigma \in \mathbb{R}, \tag{14}$$

Otherwise, if $k$ is odd, then $c = 0$ is the zero tensor.

*Case $p = -1$:* Assume $p = -1$. If $k - d$ is even and $k \geq d$, then $c$ can be written in the form

$$c = \sum_{\sigma \in S_k} \beta_\sigma \Big(\delta^{\otimes \frac{k-d}{2}} \otimes \epsilon\Big)^{\sigma} \tag{15}$$

for any $\beta_\sigma \in \mathbb{R}$. Otherwise, if $k - d$ is odd or $k < d$, then $c = 0$ is the zero tensor.

The result of Theorem 1 is a clean theoretical characterization of $O(d)$-equivariant polynomial tensor functions with arbitrary order tensor inputs. However, computing large polynomials with all possible $O(d)$-isotropic tensors is impractical. One option is considering low-degree polynomials as in Example 2. Alternatively, in many applications we only need a function that has $1_{(+)}$-tensors (i.e. vectors) as input and a $k_{(+)}$-tensor as output, and the problem takes on a form more amenable to computation.

The following corollary says that when the inputs of the $O(d)$-equivariant function are only vectors, we can write the function as a linear combination where the basis elements are permutations of the input vectors and Kronecker deltas, and the coefficients are scalar functions that only depend on the pairwise inner products of the vectors. The proofs of this corollary and Lemma 1 are in Appendix C.

**Corollary 1.** Let $f : \prod_{i=1}^{n} \mathcal{T}_1(\mathbb{R}^d, +) \to \mathcal{T}_{k'}(\mathbb{R}^d, +)$ be an $O(d)$-equivariant polynomial function. Then, we may write it as

$$f(v_1, \ldots, v_n) = \sum_{t=0}^{\lfloor \frac{k'}{2} \rfloor} \sum_{\sigma \in S_{k'}} \sum_{1 \leq J_1 \leq \ldots \leq J_{k'-2t} \leq n} q_{t,\sigma,J}\Big(\big(\langle v_i, v_j\rangle\big)_{i,j=1}^{n}\Big) \Big(v_{J_1} \otimes \ldots \otimes v_{J_{k'-2t}} \otimes \delta^{\otimes t}\Big)^{\sigma},$$

$$\tag{16}$$

where $J = (J_1, \ldots, J_{k'-2t})$, and the function $q_{t,\sigma,J}$ depends on the tuple $(t, \sigma, J)$ and is a polynomial of all $n^2$ possible inner products between the input vectors.

The second factor is a permutation of the outer product of $t$ Kronecker deltas and $k' - 2t$ of the input vectors $v_1, \ldots, v_n$, possibly with repeats. The first sum is over the possible numbers of Kronecker deltas 0 to $\lfloor \frac{k'}{2} \rfloor$, where $\lfloor \cdot \rfloor$ is the floor function. The second sum is over the possible permutations of the $k'$ axes, and the third sum is over choosing $k' - 2t$ vectors from $v_1$ to $v_n$, allowing repeated vectors.

**Remark 2.** Note that Corollary 1 characterizes polynomial functions, but if we allow the $q_{t,\sigma,J}$ to be more general (e.g. in the class of continuous or smooth functions), then we obtain a parameterization of a larger class of $O(d)$-equivariant functions. In the experiments in Section 5, we set the $q_{t,\sigma,J}$ to be learnable multi-layer perceptrons (MLPs). We are unsure if a characterization of this sort can be stated for all continuous $O(d)$-equivariant functions. However, by the Stone–Weierstrass theorem any continuous function can be approximated by a polynomial function to arbitrary accuracy on any fixed compact set, so constructing an architecture that can represent equivariant polynomial functions is sufficient to approximately represent equivariant continuous functions (see Yarotsky (2022)).

**Remark 3.** One interesting potential extension of Corollary 1 is to parameterize the polynomials $f$ that are simultaneously $O(d)$-equivariant and $S_n$-invariant, where $S_n$ is the symmetric group of order $n$ acting by permuting the inputs $v_1, \ldots, v_n$. Implementations of such models may be possible by adapting techniques from DeepSets Zaheer et al. (2017) or graph neural networks Scarselli et al. (2008). However, parameterizing all permutation invariant polynomial functions $q$ may be as hard as solving the graph isomorphism problem, which is currently intractable. Recent work gives an efficient parameterization of a class of invariant functions that is almost separating and could potentially be used to implement the $q$ functions Blum-Smith et al. (2024).

## 4 GENERALIZATIONS TO OTHER GROUPS

The results regarding $O(d)$-equivariant tensor maps from Section 3 are a particular case of a more general result involving algebraic groups. We work the full generalization in Appendix F where we give all the details of the proofs.

Recall that we can define $O(d)$ as follows:

$$O(d) := \{g \in \mathrm{GL}(\mathbb{R}^d) \mid g^\top g = \mathbb{I}_d\}, \tag{17}$$

where $\mathbb{I}$ is the identity matrix. In other words, $O(d)$ is the subgroup of linear transformations preserving the Euclidean inner product. However, in some contexts, we might be interested in preserving more general bilinear products on $\mathbb{R}^d$, such as the *Minkowski inner product*

$$\langle u, v \rangle_s := u^\top \mathbb{I}_{s,d-s} v,$$

where $\mathbb{I}_{s,d-s} := \begin{pmatrix} \mathbb{I}_s & \\ & -\mathbb{I}_{d-s} \end{pmatrix}$, or, for $d$ even, the *symplectic product*

$$\langle u, v \rangle_{\mathrm{symp}} := u^\top J_d v$$

where $J_d := \begin{pmatrix} & \mathbb{I}_{d/2} \\ -\mathbb{I}_{d/2} & \end{pmatrix}$. The subgroups of linear maps preserving these bilinear products give respectively the *indefinite orthogonal group* (which is the linear part of the *Lorentz group* when $d = 4$ and $s \in \{1, 3\}$) given by

$$O(s, d-s) := \{g \in \mathrm{GL}(\mathbb{R}^d) \mid g^\top \mathbb{I}_{s,d-s} g = \mathbb{I}_{s,d-s}\}, \tag{18}$$

and, when $d$ is even, the *symplectic group* given by

$$Sp(d) := \{g \in \mathrm{GL}(\mathbb{R}^d) \mid g^\top J_d g = J_d\}. \tag{19}$$

For any of these groups $G$, we can consider the modules $\mathcal{T}_k(\mathbb{R}^d, \chi) := (\mathbb{R}^d)^{\otimes k}$, where $\chi : G \to \mathbb{R}^*$ is an algebraic group homomorphism, where the action is given by the linear extension of

$$g \cdot (v_1 \otimes \cdots \otimes v_k) = \chi(g)(g \cdot v_1) \otimes \cdots \otimes (g \cdot v_k). \tag{20}$$

When $G = O(s, d - s)$, with $s \neq 0, d$, we have four possible $\chi$: $\chi_{+,+}$ being always equal to 1, $\chi_{+,-}$ being the sign of the determinant of the bottom-right $(d - s) \times (d - s)$ submatrix, $\chi_{-,+}$ being the sign of the determinant of the top-left $s \times s$ submatrix, and $\chi_{-,-}$ being the determinant of the matrix. Hence, we can represent them by $(p_1, p_2)$, where $p_i \in \{-1, +1\}$. When $G = Sp(d)$, we have that $\chi$ can only be the trivial group-homomorphism. (It follows, for instance, from the representation theory of simple Lie algebras from (Fulton & Harris, 2013, Part III) and a standard abelianization argument).

Additionally, we have $G$-equivariant contractions $\iota_k^G : \mathcal{T}_{2k+k'}(\mathbb{R}^d, \chi) \to \mathcal{T}_{2k+k'}(\mathbb{R}^d, \chi)$ given by

$$\iota_k^{O(s,d-s)}(a) := [a]_{i_1,\ldots,i_k,j_1,\ldots,j_k,\ell_1,\ldots,\ell_{k'}} [\mathbb{I}_{s,d-s}]_{i_1,j_1} \cdots [\mathbb{I}_{s,d-s}]_{i_k,j_k} \tag{21}$$

and

$$\iota_k^{Sp(d)}(a) := [a]_{i_1,\ldots,i_k,j_1,\ldots,j_k,\ell_1,\ldots,\ell_{k'}} [J_d]_{i_1,j_1} \cdots [J_d]_{i_k,j_k} . \tag{22}$$

Under these notations, we can state the generalization of Theorem 1 as follows. Recall that an *entire function* is a function that is analytic and whose Taylor series converges globally at any point.

**Theorem 2.** Let $G$ be either $O(s, d - s)$ or $Sp(d)$ and $f : \prod_{i=1}^n \mathcal{T}_{k_i}(\mathbb{R}^d, \chi_i) \to \mathcal{T}_{k'}(\mathbb{R}^d, \chi')$ be a $G$-equivariant entire function. Then we may write $f$ as follows:

$$f(a_1, \ldots, a_n) = \sum_{r=0}^{\infty} \sum_{1 \leq \ell_1 \leq \cdots \leq \ell_r \leq n} \iota_{k_{\ell_1,\ldots,\ell_r}}^G (a_{\ell_1} \otimes \ldots \otimes a_{\ell_r} \otimes c_{\ell_1,\ldots,\ell_r}) \tag{23}$$

where $c_{\ell_1,\ldots,\ell_r} \in \mathcal{T}_{k_{\ell_1,\ldots,\ell_r}+k'}(\mathbb{R}^d, \chi_{\ell_1,\ldots,\ell_r} \chi')$ is a $G$-isotropic tensor, i.e., a tensor in $\mathcal{T}_{k_{\ell_1,\ldots,\ell_r}+k'}(\mathbb{R}^d, \chi_{\ell_1,\ldots,\ell_r} \chi')$ invariant under the action of $G$; for $k_{\ell_1,\ldots,\ell_r} := \sum_{q=1}^r k_{\ell_q}$ and $\chi_{\ell_1,\ldots,\ell_r} = \prod_{q=1}^r \chi_{\ell_q}$.

Using the above theorem and an analogous version of Lemma 1 ((Roe Goodman, 2009, Theorem 5.3.3), see Proposition 7 in Appendix F), we can then prove the following corollary, which generalizes Corollary 1.

**Corollary 2.** Let $G$ be either $O(s, d - s)$ or $Sp(d)$ and $f : \prod_{i=1}^n \mathcal{T}_1(\mathbb{R}^d, \chi_0) \to \mathcal{T}_k(\mathbb{R}^d, \chi_0)$, with $\chi_0$ the constant map to 1, be a $G$-equivariant entire function. Then we may write $f$ as follows:

$$f(v_1, \ldots, v_n) = \sum_{t=0}^{\lfloor \frac{k}{2} \rfloor} \sum_{\sigma \in S_k} \sum_{1 \leq J_1 \leq \cdots \leq J_{k-2t} \leq n} q_{t,\sigma,J}\left((\langle v_i, v_j \rangle_G)_{i,j=1}^n\right) \left(v_{J_1} \otimes \ldots \otimes v_{J_{k-2t}} \otimes \theta_G^{\otimes t}\right)^{\sigma} \tag{24}$$

where $\langle \cdot, \cdot \rangle_G = \langle \cdot, \cdot \rangle_s$ and $\theta_G = [\mathbb{I}_{s,d-s}]_{i,j}$ if $G = O(s, d - s)$, and $\langle \cdot, \cdot \rangle_G = \langle \cdot, \cdot \rangle_{\text{symp}}$ and $\theta_G = [J_d]_{i,j}$ if $G = Sp(d)$, and $q_{t,\sigma,J}$ is an entire function that depends on the tuple $(t, \sigma, J)$ and whose inputs are all possible inner products between the input vectors and whose output is a scalar.

## 5 NUMERICAL EXPERIMENTS

With the preceding theory in place, we can define a machine learning model to learn a new algorithm of interest. The sum of squares solution to the sparse vector recovery problem reduces to finding the optimal $O(d)$-equivariant function with vector inputs and a $2_{(+)}$-tensor output, and the form of this function is given by Corollary 1. The learned algorithms outperform state-of-the-art methods for this problem in several regimes, and operate in settings where theoretical guarantees have yet to be developed. As such, they may point towards conjectures for more general classes of spectral algorithms. The code is open-source and will be released after anonymous review.

### 5.1 PROBLEM SETUP

We consider the problem of finding a planted sparse vector in a linear subspace. This problem was introduced by Spielman, Wang, and Wright in Spielman et al. (2012) in the context of dictionary learning. It was further studied in Hopkins et al. (2016) and Mao & Wein (2022).

**Problem 1.** Let $v \in \mathbb{R}^n$ be an (approximately) sparse vector of unit length. Construct $v_0, \ldots, v_{d-1} \in \mathbb{R}^n$ from $v$ using some noise strategy. We consider $S$ to be an $n \times d$ matrix whose columns form an orthonormal basis of span$\{v_0, \ldots, v_{d-1}\}$. The problem is to recover $v$ from $S$.

The methods developed in Hopkins et al. (2016) and Mao & Wein (2022) assume that for sparsity parameter $\varepsilon \le 1/3$, $\|v\|_4^4 \ge \frac{1}{\varepsilon n}$ and that $v_0 = v$ and $v_1, \ldots, v_{d-1} \sim \mathcal{N}\left(\mathbf{0}_n, \frac{1}{n}\mathbb{I}_n\right)$. We will violate some or all of these assumptions in our experiments on synthetic data and MNIST digits.

In synthetic data experiments, we sample $v$ using one of four methods: Accept/Reject (AR), Bernoulli-Gaussian (BG), Corrected Bernoulli-Gaussian (CBG), and Bernoulli-Rademacher (BR). Vectors sampled using AR are rejected if they do not satisfy $\|v\|_4^4 \ge \frac{1}{\varepsilon n}$, while vectors sampled by the other methods only satisfy that bound in expectation (Appendix I.1). We set $v_0 = v$ and sample $v_1, \ldots, v_{d-1} \sim \mathcal{N}(\mathbf{0}_n, \Sigma)$ where $\Sigma$ can be the identity, a non-identity diagonal covariance, or a random covariance from a Wishart distribution (Appendix I.2). We then get a random orthonormal basis of $v_0, \ldots, v_{d-1}$ (Appendix I.4).

For experiments on more realistic data, we turn to MNIST. Since the background of each image consists of 0 values and the digit of nonzero values is on average 20% of the pixels, we can consider each MNIST image as our sparse target vector $v \in \mathbb{R}^n$, where $n = 28^2$. We construct a subspace from several noisy copies, and Problem 1 is equivalent to recovering the original, denoised image from the noisy copies. We stress that this is an illustrative example of the sparse vector recovery problem; we make no claims that this is a state-of-the-art method for the image denoising problem in general.

To construct $v_0, \ldots, v_{d-1}$ for the MNIST data, we use four different noise strategies. In the first noise scheme, the Random Subspace (RND), we let $v_0 = v$ and $v_1, \ldots, v_{d-1} \sim \mathcal{N}(\mathbf{0}_n, \mathbb{I}_n)$ so that this method is directly comparable to the synthetic data experiments above. In the next three noise schemes, each $v_i$, *including* $v_0$, is a noisy copy of $v$ so that $v$ is not necessarily in the subspace of $v_0, \ldots, v_{d-1}$. We use additive Gaussian noise (GAU), Bernoulli pixel noise (BER), and a random block mask (BLK). See Figure 1 for a depiction of the noise and I.3 for further details.

## 5.2 Models

We consider a matrix $S \in \mathbb{R}^{n \times d}$ whose columns are an orthonormal basis for $\text{span}\{v_0, \ldots, v_{d-1}\}$ and denote its rows by $a_1^\top, \ldots a_n^\top \in \mathbb{R}^d$. Let $\mathcal{S}_d$ be the space of $d \times d$ symmetric matrices. We consider $h : (\mathbb{R}^d)^n \to \mathcal{S}_d$ and let $\lambda_{\text{vec}} : \mathcal{S}_d \to \mathbb{R}^d$ be the function that takes a symmetric matrix as input and outputs a normalized eigenvector corresponding to the top eigenvalue. Then, the estimate of the planted sparse vector will be given by

$$\hat{v} = S\,\lambda_{\text{vec}}(h(a_1, \ldots, a_n))\,, \tag{25}$$

for an appropriate $h$ that can be learned from data or derived by other means.

Since $S$ may be any orthogonal basis of $\text{span}\{v_0, \ldots, v_{d-1}\}$, we would like (25) to be invariant to $O(d)$. It is sufficient for $h$ to be $O(d)$-equivariant to guarantee this invariance – see Appendix G.

The models we consider differ in their choice of function $h$. In Hopkins et al. (2016) (SOS-I), the function $h$ is

$$h(a_1, \ldots, a_n) := \sum_{i=1}^n \left( \|a_i\|_2^2 - \frac{d}{n} \right) a_i a_i^\top \,, \tag{26}$$

and in Mao & Wein (2022) (SOS-II) $h$ is defined as

$$h(a_1, \ldots, a_n) := \sum_{i=1}^n \left( \|a_i\|_2^2 - \frac{d-1}{n} \right) a_i a_i^\top - \frac{3}{n}\mathbb{I}_n \,. \tag{27}$$

Both models are proven to recover the planted sparse vector under different sampling assumptions described in Appendix I.1. Note that equations (26) and (27) are $O(d)$-equivariant and a special case of Corollary 1 since they define a sum of outer products of the inputs with coefficients that are polynomial functions of inner products of the inputs.

In comparison to these fixed methods, we propose two machine learning-based models defined using the results of Section 3. The first model, SparseVectorHunter (SVH), will parameterize

$$h(a_1, \ldots, a_n) = \left[ \sum_{i=1}^n \sum_{j=i}^n q_{i,j}\Big( (\langle a_\ell, a_m \rangle)_{\ell,m=1}^n \Big) \frac{1}{2} \big( a_i a_j^\top + a_j a_i^\top \big) \right] + q_{\mathbb{I}}\Big( (\langle a_\ell, a_m \rangle)_{\ell,m=1}^n \Big)\mathbb{I}_d \,, \tag{28}$$

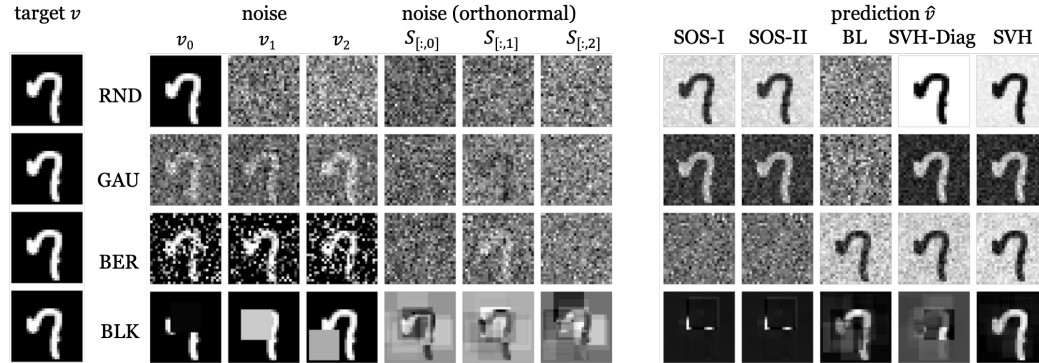

Figure 1: MNIST target $v$ (left), noise inputs from different schemes per row (middle), and the corresponding model predictions $\hat{v}$ (right): Our methods (SVH-Diag, SVH) outperform existing sum-of-squares methods (SOS-I, SOS-II) and the non-equivariant baseline (BL) across all noise schemes including Random Subspace (RND), Gaussian noise (GAU), Bernoulli noise (BER), and Block Noise (BLK).

and the second model, SparseVectorHunterDiagonal (SVH-Diag), will parameterize

$$h(a_1, \ldots, a_n) = \left[ \sum_i^n q_i \left( \left( \|a_\ell\|_2^2 \right)_{\ell=1}^n \right) a_i a_i^\top \right] + q_\mathbb{I} \left( \left( \|a_\ell\|_2^2 \right)_{\ell=1}^n \right) \mathbb{I}_d \,, \tag{29}$$

where $q_{i,j}$, $q_i$, and $q_\mathbb{I}$ are $O(d)$-invariant scalar functions. The form of equation (28) follows from Corollary 1 as shown in Appendix G. By averaging the general form of a matrix valued $O(d)$-equivariant function with its transpose, we obtain the form of any $O(d)$-equivariant polynomial function that outputs a symmetric matrix. Equation (29) follows the scheme of (28) but only includes inner and outer products of the same vectors to be more directly comparable to (26) and (27). Corollary 1 specifies that $q_{i,j}$, $q_i$, and $q_\mathbb{I}$ should be polynomials, but we will approximate them with dense neural networks. The networks themselves are multi-layer perceptrons (MLP) with 2 hidden layers, width of 128, and ReLU activation functions.

To demonstrate the benefits of equivariance, we also implement a non-equivariant baseline model (BL) which takes as input the $nd$ components of $S$ and outputs the $d + \binom{d}{2}$ components of a symmetric $d \times d$ matrix. This is implemented as a multi-layer perceptron with 2 hidden layers, width of 128, and ReLU activation functions to match the architecture of the SVH. For training details on all these models, see Appendix I.5.

## 5.3 Results

The synthetic data results on the test data are displayed in Table 1. The SOS-I and SOS-II methods perform best when their assumptions are met, such as identity covariance for the noise vectors, but perform worse than the SVH models when using Random or Diagonal covariance. One exception to this trend is the Bernoulli-Gaussian sampling for the sparse vectors. This is likely because if $v \sim \text{BG}$, then $\mathbb{E}\left[ \|v\|_4^4 \right] = \frac{3}{\varepsilon n}$ (Appendix I.1). This is significantly above the required sparsity condition of $\|v\|_4^4 \geq \frac{1}{\varepsilon n}$, so the vectors are more sparse and therefore easier to estimate. By contrast, the Corrected Bernoulli-Gaussian has $\mathbb{E}\left[ \|v\|_4^4 \right] = \frac{1}{\varepsilon n}$ (Appendix I.1) and we once again see that the learned SVH models perform better when we can't rely on stringent data assumptions.

This story is even more pronounced in the MNIST data experiments, shown in Figure 1 and Table 2. The learned SVH models outperform the SOS methods across every noise type for both $d = 10$ and $d = 20$. The Gaussian, Bernoulli, and Block noise schemes break almost all of the assumptions of the SOS methods, including the fact that the sparse vector is not explicitly in the subspace $v_0, \ldots, v_{d-1}$. The block noise scheme, essentially an image inpainting task, is particularly egregious because not only is the noise in nearby pixels highly correlated, the noise itself is sparse in a sense. The learned models are able to adapt to all these new settings and perform well.

| sampling | $\Sigma$ | SOS-I | SOS-II | BL | SVH-Diag | SVH |
|---|---|---|---|---|---|---|
| A/R | Random | $0.610 \pm 0.009$ | $0.610 \pm 0.009$ | $0.241 \pm 0.019$ | $0.493 \pm 0.005$ | $\mathbf{0.938 \pm 0.002}$ |
| | Diagonal | $0.448 \pm 0.012$ | $0.448 \pm 0.012$ | $0.196 \pm 0.011$ | $\mathbf{0.589 \pm 0.026}$ | $0.465 \pm 0.027$ |
| | Identity | $\mathbf{0.606 \pm 0.014}$ | $\mathbf{0.606 \pm 0.014}$ | $0.196 \pm 0.008$ | $0.351 \pm 0.065$ | $0.190 \pm 0.008$ |
| BG | Random | $\mathbf{0.962 \pm 0.002}$ | $\mathbf{0.962 \pm 0.002}$ | $0.242 \pm 0.006$ | $0.917 \pm 0.004$ | $0.937 \pm 0.002$ |
| | Diagonal | $\mathbf{0.949 \pm 0.005}$ | $\mathbf{0.949 \pm 0.005}$ | $0.205 \pm 0.013$ | $0.914 \pm 0.006$ | $0.463 \pm 0.018$ |
| | Identity | $\mathbf{0.962 \pm 0.002}$ | $\mathbf{0.962 \pm 0.002}$ | $0.196 \pm 0.009$ | $0.908 \pm 0.006$ | $0.342 \pm 0.043$ |
| CBG | Random | $0.412 \pm 0.017$ | $0.412 \pm 0.017$ | $0.239 \pm 0.012$ | $0.372 \pm 0.011$ | $\mathbf{0.935 \pm 0.002}$ |
| | Diagonal | $0.288 \pm 0.018$ | $0.288 \pm 0.018$ | $0.206 \pm 0.003$ | $\mathbf{0.550 \pm 0.026}$ | $0.460 \pm 0.022$ |
| | Identity | $\mathbf{0.412 \pm 0.011}$ | $\mathbf{0.412 \pm 0.011}$ | $0.198 \pm 0.005$ | $0.239 \pm 0.025$ | $0.197 \pm 0.011$ |
| BR | Random | $0.526 \pm 0.020$ | $0.526 \pm 0.020$ | $0.923 \pm 0.004$ | $0.437 \pm 0.034$ | $\mathbf{0.957 \pm 0.001}$ |
| | Diagonal | $0.334 \pm 0.024$ | $0.334 \pm 0.024$ | $0.864 \pm 0.005$ | $0.588 \pm 0.011$ | $\mathbf{0.903 \pm 0.004}$ |
| | Identity | $0.524 \pm 0.010$ | $0.524 \pm 0.010$ | $0.845 \pm 0.006$ | $0.317 \pm 0.046$ | $\mathbf{0.889 \pm 0.003}$ |

Table 1: Test error comparison on synthetic data averaged over 5 trials ($n = 100, d = 5, \epsilon = 0.25$) with the standard deviation given by $\pm 0.xxx$. The metric $\langle v, \hat{v} \rangle^2$ ranges from 0 to 1 with 1 indicating the estimate $\hat{v}$ identical to the true $v$. For each row, the best value is **bolded**.

| | Noise Type | SOS-I | SOS-II | BL | SVH-Diag | SVH |
|---|---|---|---|---|---|---|
| $d = 10$ | Random | $0.966 \pm 0.003$ | $0.966 \pm 0.003$ | $0.509 \pm 0.086$ | $\mathbf{1.000 \pm 0.000}$ | $0.993 \pm 0.000$ |
| | Gaussian | $0.776 \pm 0.005$ | $0.776 \pm 0.005$ | $0.632 \pm 0.009$ | $\mathbf{0.835 \pm 0.001}$ | $0.829 \pm 0.001$ |
| | Bernoulli | $0.091 \pm 0.006$ | $0.091 \pm 0.006$ | $0.680 \pm 0.004$ | $\mathbf{0.745 \pm 0.001}$ | $0.743 \pm 0.001$ |
| | Block | $0.096 \pm 0.012$ | $0.096 \pm 0.012$ | $0.623 \pm 0.020$ | $0.451 \pm 0.014$ | $\mathbf{0.872 \pm 0.006}$ |
| $d = 20$ | Random | $0.886 \pm 0.004$ | $0.886 \pm 0.004$ | $0.061 \pm 0.013$ | $\mathbf{1.000 \pm 0.000}$ | $0.991 \pm 0.000$ |
| | Gaussian | $0.749 \pm 0.005$ | $0.749 \pm 0.005$ | $0.448 \pm 0.034$ | $\mathbf{0.910 \pm 0.000}$ | $0.903 \pm 0.000$ |
| | Bernoulli | $0.031 \pm 0.004$ | $0.031 \pm 0.004$ | $0.635 \pm 0.007$ | $\mathbf{0.774 \pm 0.000}$ | $0.773 \pm 0.000$ |
| | Block | $0.069 \pm 0.006$ | $0.069 \pm 0.006$ | $0.556 \pm 0.023$ | $0.440 \pm 0.026$ | $\mathbf{0.917 \pm 0.004}$ |

Table 2: MNIST test performance averaged over 5 trials with the standard deviation $\pm 0.xxx$. The metric is $\langle v_0, \hat{v} \rangle^2$, ranging from 0 to 1, with values closer to 1 meaning that the vectors are closer. For each row, the best value is **bolded**.

Finally, we see that in all experiments, the baseline learned model generalizes poorly, despite doing well on the training data (Table 4 in Appendix). This is consistent with the claim that enforcing symmetries improves generalization performance Bietti et al. (2021); Petrache & Trivedi (2024); Elesedy (2021a).

# 6 DISCUSSION

This paper provides a characterization of polynomial functions from multiple tensor inputs to tensor outputs that are equivariant with respect to the diagonal action by classical Lie groups, including the orthogonal group, the symplectic group, and the Lorentz group.

Our main goal behind this characterization is to define equivariant machine learning models. We applied the resulting models to learning algorithms for the sparse vector recovery problem. In the spirit of the neural algorithmic reasoning framework Veličković & Blundell (2021), the proposed machine learning methods are "aligned" to known algorithms with provable performance guarantees. The learned algorithms outperform state-of-the-art algorithms for this problem in several regimes, including those where there are no known algorithms with theoretical guarantees.

The application of our characterization is especially useful when the input consists of $1_{(+)}$-tensors, and the output is a $k_{(+)}$-tensor. For these problems, a parameterization based on invariant scalars, similar to Villar et al. (2021), is available and easy to implement. For problems where the input tensors have higher order, the implementation would be less efficient. We also assume no additional structure on the relationship between the input tensors; in those situations, other techniques may be required.

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

## A   BASIC PROPERTIES OF $O(d)$ ACTIONS ON TENSORS

In this section, we will show that the basic operations are $O(d)$-equivariant and linear by direct computation. We do so explicitly by performing routine computations. However, the universal property of tensor products, which we use in Appendix F, would give immediate proofs of these statements.

**Proposition 1.** The outer product is a $O(d)$-equivariant bilinear map. In other words, for $g \in O(d)$, $a, a' \in \mathcal{T}_k(\mathbb{R}^d, p)$, $b, b' \in \mathcal{T}_{k'}(\mathbb{R}^d, p')$ and $\alpha, \beta \in \mathbb{R}$, we have $g \cdot (a \otimes b) = (g \cdot a) \otimes (g \cdot b)$, $(\alpha a + \beta a') \otimes b = \alpha(a \otimes b) + \beta(a' \otimes b)$, and $a \otimes (\alpha b + \beta b') = \alpha(a \otimes b) + \beta(a \otimes b')$. In particular, if $c \in \mathcal{T}_{k'}(\mathbb{R}^d, p')$ is an $O(d)$-isotropic tensor, then the function mapping

$$\mathcal{T}_k(\mathbb{R}^d, p) \to \mathcal{T}_{k+k'}(\mathbb{R}^d, pp') \quad \text{by} \quad a \mapsto a \otimes c \tag{30}$$

is an $O(d)$-equivariant linear map.

**Proposition 2.** The $k$-contraction $\iota_k : \mathcal{T}_{2k+k'}(\mathbb{R}^d, p) \to \mathcal{T}_{k'}(\mathbb{R}^d, p)$, see Definition 4, is an $O(d)$-equivariant linear map.

**Proposition 3.** For fixed $\sigma \in S_k$, the tensor index permutation mapping $\mathcal{T}_k(\mathbb{R}^d, p) \to \mathcal{T}_k(\mathbb{R}^d, p)$ by $a \mapsto a^\sigma$ is an $O(d)$-equivariant linear map.

*Proof of Proposition 1.* First, we establish equivariance. Let $a \in \mathcal{T}_k(\mathbb{R}^d, p)$, $b \in \mathcal{T}_{k'}(\mathbb{R}^d, p')$, and $g \in O(d)$. We have

$$[g \cdot (a \otimes b)]_{j_1, \ldots, j_{k+k'}}$$

$$= \det(M(g))^{\frac{1 - p\, p'}{2}} [(a \otimes b)]_{i_1, \ldots, i_{k+k'}} [M(g)]_{j_1, i_1} \cdots [M(g)]_{j_{k+k'}, i_{k+k'}}$$

$$= \det(M(g))^{\frac{1 - p}{2}} \det(M(g))^{\frac{1 - p'}{2}} [a]_{i_1, \ldots, i_k} [b]_{i_{k+1}, \ldots, i_{k+k'}} [M(g)]_{j_1, i_1} \cdots$$
$$\cdots [M(g)]_{j_k, i_k} [M(g)]_{j_{k+1}, i_{k+1}} \cdots [M(g)]_{j_{k+k'}, i_{k+k'}}$$

$$= \left( \det(M(g))^{\frac{1 - p}{2}} [a]_{i_1, \ldots, i_k} [M(g)]_{j_1, i_1} \cdots [M(g)]_{j_k, i_k} \right)$$

$$\left( \det(M(g))^{\frac{1 - p'}{2}} [b]_{i_{k+1}, \ldots, i_{k+k'}} [M(g)]_{j_{k+1}, i_{k+1}} \cdots [M(g)]_{j_{k+k'}, i_{k+k'}} \right)$$

$$= [g \cdot a]_{j_1, \ldots, j_k} [g \cdot b]_{j_{k+1}, \ldots, j_{k+k'}}$$

$$= [g \cdot a \otimes g \cdot b]_{j_1, \ldots, j_{k+k'}},$$

where the second equality uses the fact that

$$\det(M(g))^{\frac{1 - p\, p'}{2}} = \det(M(g))^{\frac{1 - p}{2}} \det(M(g))^{\frac{1 - p'}{2}},$$

which is straightforward to verify via a case analysis over possible parameter values (i.e., $p, p' \in \{+1, -1\}$ and $\det(M(g)) \in \{+1, -1\}$).

Next, we verify linearity. Let $a, a' \in \mathcal{T}_k(\mathbb{R}^d, p)$, $b \in \mathcal{T}_{k'}(\mathbb{R}^d, p')$, and $\alpha, \beta \in \mathbb{R}$. Then,

$$[(\alpha a + \beta a') \otimes b]_{i_1, \ldots, i_{k+k'}} = [(\alpha a + \beta a')]_{i_1, \ldots, i_k} [b]_{i_{k+1}, \ldots, i_{k+k'}}$$

$$= \alpha [a]_{i_1, \ldots, i_k} [b]_{i_{k+1}, \ldots, i_{k+k'}} + \beta [a']_{i_1, \ldots, i_k} [b]_{i_{k+1}, \ldots, i_{k+k'}}$$

$$= \alpha [a \otimes b]_{i_1, \ldots, i_{k+k'}} + \beta [a' \otimes b]_{i_1, \ldots, i_{k+k'}}.$$

The linearity in the second argument follows in the same manner.

Finally, (30) follows immediately from the fact that the bilinear $O(d)$-equivariance. $\qquad\square$

*Proof of Proposition 2.* To establish equivariance, let $a \in \mathcal{T}_{2k+k'}(\mathbb{R}^d, p)$ and let $g \in O(d)$. Then,

$$[g \cdot \iota_k(a)]_{j_1,\ldots,j_{k'}}$$
$$= \det(M(g))^{\frac{1-p}{2}} [a]_{\ell_1,\ldots,\ell_k,\ell_1,\ldots,\ell_k,i_1,\ldots,i_{k'}} [M(g)]_{j_1,i_1} \cdots [M(g)]_{j_{k'},i_{k'}}$$
$$= \det(M(g))^{\frac{1-p}{2}} [a]_{\ell_1,\ldots,\ell_{2k},i_1,\ldots,i_{k'}} [\delta]_{\ell_1,\ell_{k+1}} \cdots [\delta]_{\ell_k,\ell_{2k}} [M(g)]_{j_1,i_1} \cdots [M(g)]_{j_{k'},i_{k'}}$$
$$= \det(M(g))^{\frac{1-p}{2}} [a]_{\ell_1,\ldots,\ell_{2k},i_1,\ldots,i_{k'}} [M(g)]_{\ell_1,m_1} [M(g)]_{\ell_{k+1},m_1} \cdots$$
$$\qquad \cdots [M(g)]_{\ell_k,m_k} [M(g)]_{\ell_{2k},m_k} [M(g)]_{j_1,i_1} \cdots [M(g)]_{j_{k'},i_{k'}}$$
$$= [g \cdot a]_{m_1,\ldots,m_k,m_1,\ldots,m_k,j_1,\ldots,j_{k'}}$$
$$= [\iota_k(g \cdot a)]_{j_1,\ldots,j_{k'}},$$

where the third equality uses the fact that $\delta = M(g) M(g)^\top$.

Next, to establish linearity, let $a, b \in \mathcal{T}_{2k+k'}(\mathbb{R}^d, p)$ and let $\alpha, \beta \in \mathbb{R}$. Then,

$$[\iota_k(\alpha a + \beta b)]_{j_1,\ldots,j_{k'}} = [\alpha a + \beta b]_{i_1,\ldots,i_k,i_1,\ldots,i_k,j_1,\ldots,j_{k'}}$$
$$= \alpha [a]_{i_1,\ldots,i_k,i_1,\ldots,i_k,j_1,\ldots,j_{k'}} + \beta [b]_{i_1,\ldots,i_k,i_1,\ldots,i_k,j_1,\ldots,j_{k'}}$$
$$= \alpha [\iota_k(a)]_{j_1,\ldots,j_{k'}} + \beta [\iota_k(b)]_{j_1,\ldots,j_{k'}}.$$

This completes the proof. $\qquad\square$

*Proof of Proposition 3.* Fix $\sigma \in S_k$. To establish equivariance, let $a \in \mathcal{T}_k(\mathbb{R}^d, p)$ and $g \in O(d)$. Then,

$$[g \cdot (a^\sigma)]_{j_1,\ldots,j_k} = \det(M(g))^{\frac{1-p}{2}} [a^\sigma]_{i_1,\ldots,i_k} [M(g)]_{j_1,i_1} \cdots [M(g)]_{j_k,i_k}$$
$$= \det(M(g))^{\frac{1-p}{2}} [a]_{i_{\sigma^{-1}(1)},\ldots,i_{\sigma^{-1}(k)}} [M(g)]_{j_1,i_1} \cdots [M(g)]_{j_k,i_k}$$
$$= \det(M(g))^{\frac{1-p}{2}} [a]_{i_{\sigma^{-1}(1)},\ldots,i_{\sigma^{-1}(k)}} [M(g)]_{j_{\sigma^{-1}(1)},i_{\sigma^{-1}(1)}} \cdots [M(g)]_{j_{\sigma^{-1}(k)},i_{\sigma^{-1}(k)}}$$
$$= [g \cdot a]_{j_{\sigma^{-1}(1)},\ldots,j_{\sigma^{-1}(k)}}$$
$$= [(g \cdot a)^\sigma]_{j_1,\ldots,j_k},$$

where the third equality holds since we are merely reordering the $M(g)$ components—which is allowed because they are scalars.

To show linearity, let $a, b \in \mathcal{T}_k(\mathbb{R}^d, p)$ and $\alpha, \beta \in \mathbb{R}$. We have

$$[(\alpha a + \beta b)^\sigma]_{i_1,\ldots,i_k} = [\alpha a + \beta b]_{i_{\sigma^{-1}(1)},\ldots,i_{\sigma^{-1}(k)}}$$
$$= \alpha [a]_{i_{\sigma^{-1}(1)},\ldots,i_{\sigma^{-1}(k)}} + \beta [b]_{i_{\sigma^{-1}(1)},\ldots,i_{\sigma^{-1}(k)}}$$
$$= \alpha [a^\sigma]_{i_1,\ldots,i_k} + \beta [b^\sigma]_{i_1,\ldots,i_k}.$$

$\qquad\square$

## B  PROOF OF THEOREM 1

The main idea of the proof of Theorem 1 is to write out the polynomial $f$ in a way that takes advantage of the tensor operations of Section 2, then show that each term must be $O(d)$-equivariant (Lemmas 2 and 3). We then use a group averaging argument to show that $c_{\ell_1,\ldots,\ell_r}$ can be written as an $O(d)$-isotropic tensor. We state the lemmas, prove the theorem, then prove the lemmas.

**Lemma 2.** Let $f : \prod_{i=1}^n \mathcal{T}_{k_i}(\mathbb{R}^d, p_i) \to \mathcal{T}_{k'}(\mathbb{R}^d, p')$ be a polynomial map of degree $R$, and write

$$f(a_1,\ldots,a_n) = \sum_{r=0}^R f_r(a_1,\ldots,a_n),$$

where $f_r : \prod_{i=1}^n \mathcal{T}_{k_i}(\mathbb{R}^d, p_i) \to \mathcal{T}_{k'}(\mathbb{R}^d, p')$ is homogeneous degree $r$ polynomial. If $f$ is $O(d)$-equivariant, then each $f_r$ is $O(d)$-equivariant.

**Lemma 3.** Let $f_r : \prod_{i=1}^n \mathcal{T}_{k_i}(\mathbb{R}^d, p_i) \to \mathcal{T}_{k'}(\mathbb{R}^d, p')$ be a homogeneous polynomial of degree $r$. Then, we can write $f_r$ as

$$f_r(a_1, \ldots, a_n) = \sum_{1 \le \ell_1 \le \ldots \le \ell_r \le n} f_{\ell_1, \ldots, \ell_r}(a_{\ell_1}, \ldots, a_{\ell_r}), \tag{31}$$

where $f_{\ell_1, \ldots, \ell_r} : \prod_{i=1}^r \mathcal{T}_{k_{\ell_i}}(\mathbb{R}^d, p_{\ell_i}) \to \mathcal{T}_{k'}(\mathbb{R}^d, p')$ is the composition of the map

$$\prod_{i=1}^r \mathcal{T}_{k_{\ell_i}}(\mathbb{R}^d, p_{\ell_i}) \to \mathcal{T}_{\sum_{i=1}^r k_{\ell_i}}\left(\mathbb{R}^d, \prod_{i=1}^r p_{\ell_i}\right)$$

$$(a_{\ell_1}, \ldots, a_{\ell_r}) \mapsto a_{\ell_1} \otimes \ldots \otimes a_{\ell_r}$$

with a linear map $\mathcal{T}_{\sum_{i=1}^r k_{\ell_i}}(\mathbb{R}^d, \prod_{i=1}^r p_{\ell_i}) \to \mathcal{T}_{k'}(\mathbb{R}^d, p')$.

Moreover, if $f_r$ is $O(d)$-equivariant, then so are the $f_{\ell_1, \ldots, \ell_r}$.

**Remark 4.** Note that Lemma 3 is nothing more than the decomposition of $f_r$ as a sum of multihomogeneous maps in the inputs $a_1, \ldots, a_n$.

*Proof of Theorem 1.* Combining Lemmas 2 and 3, we can write $f$ as follows:

$$f(a_1, \ldots, a_n) = \sum_{r=0}^R \sum_{1 \le \ell_1 \le \ldots \le \ell_r \le n} f_{\ell_1, \ldots, \ell_r}(a_{\ell_1}, \ldots, a_{\ell_r}), \tag{32}$$

where the $f_{\ell_1, \ldots, \ell_r}$ is the composition of a linear map $\mathcal{T}_{k_{\ell_1, \ldots, \ell_r}}(\mathbb{R}^d, p_{\ell_1, \ldots, \ell_r}) \to \mathcal{T}_{k'}(\mathbb{R}^d, p')$ with the map $(a_1, \ldots, a_\ell) \mapsto a_{\ell_1} \otimes \cdots \otimes a_{\ell_r}$. Recall that $k_{\ell_1, \ldots, \ell_r} = \sum_{q=1}^r k_{\ell_q}$ and $p_{\ell_1, \ldots, \ell_r} = \prod_{q=1}^r p_{\ell_q}$. Moreover, by the lemmas, each $f_{\ell_1, \ldots, \ell_r}$ is $O(d)$-equivariant. Hence, without loss of generality, it is enough to prove the theorem in the special case

$$f(a_1, \ldots, a_n) = \lambda(a_{\ell_1} \otimes \cdots \otimes a_{\ell_r}), \tag{33}$$

where $\lambda : \mathcal{T}_{\sum_{i=1}^r k_{\ell_i}}(\mathbb{R}^d, \prod_{i=1}^r p_{\ell_i}) \to \mathcal{T}_{k'}(\mathbb{R}^d, p')$ is linear.

Now, in coordinates, we can write this map as

$$[f(a_1, \ldots, a_n)]_{j_1, \ldots, j_{k'}} = \lambda_{i_1, \ldots, i_{k_{\ell_1, \ldots, \ell_r}}, j_1, \ldots, j_{k'}}[a_{\ell_1} \otimes \cdots \otimes a_{\ell_r}]_{i_1, \ldots, i_{k_{\ell_1, \ldots, \ell_r}}}. \tag{34}$$

Consider now the tensor $c \in \mathcal{T}_{k_{\ell_1, \ldots, \ell_r} + k'}(\mathbb{R}^d, p_{\ell_1, \ldots, \ell_r} p')$ given by

$$[c]_{i_1, \ldots, i_{k_{\ell_1, \ldots, \ell_r} + k'}} = \lambda_{i_1, \ldots, i_{k_{\ell_1, \ldots, \ell_r}}, i_{k_{\ell_1, \ldots, \ell_r} + 1}, \ldots, i_{k_{\ell_1, \ldots, \ell_r} + k'}} \tag{35}$$

Then we have that

$$[f(a_1, \ldots, a_n)]_{j_1, \ldots, j_{k'}} = [c]_{i_1, \ldots, i_{k_{\ell_1, \ldots, \ell_r}}, j_1, \ldots, j_{k'}}[a_{\ell_1} \otimes \cdots \otimes a_{\ell_r}]_{i_1, \ldots, i_{k_{\ell_1, \ldots, \ell_r}}} \tag{36}$$

$$= [a_{\ell_1} \otimes \cdots \otimes a_{\ell_r} \otimes c]_{i_1, \ldots, i_{k_{\ell_1, \ldots, \ell_r}}, i_1, \ldots, i_{k_{\ell_1, \ldots, \ell_r}}, j_1, \ldots, j_{k'}} \tag{37}$$

$$= [\iota_{k_{\ell_1, \ldots, \ell_r}}(a_{\ell_1} \otimes \cdots \otimes a_{\ell_r} \otimes c)]_{j_1, \ldots, j_{k'}}, \tag{38}$$

after using the definition of $k$-contraction. Hence

$$f(a_1, \ldots, a_n) = \iota_{k_{\ell_1, \ldots, \ell_r}}(a_{\ell_1} \otimes \cdots \otimes a_{\ell_r} \otimes c). \tag{39}$$

Since $f$ is $O(d)$-equivariant, we have that for all $g \in O(d)$,

$$f(a_1, \ldots, a_n) = \iota_{k_{\ell_1, \ldots, \ell_r}}(a_{\ell_1} \otimes \cdots \otimes a_{\ell_r} \otimes g \cdot c). \tag{40}$$

To see this, we argue as follows:

$$f(a_1, \ldots, a_n)$$

$$= f(g \cdot (g^{-1} \cdot a_1), \ldots, g \cdot (g^{-1} \cdot a_n))$$

$$= g \cdot f((g^{-1} \cdot a_1), \ldots, (g^{-1} \cdot a_n)) \qquad (f \ O(d)\text{-equivariant})$$

$$= g \cdot \iota_{k_{\ell_1, \ldots, \ell_r}}((g^{-1} \cdot a_{\ell_1}) \otimes \cdots \otimes (g^{-1} \cdot a_{\ell_r}) \otimes c)$$

$$= \iota_{k_{\ell_1, \ldots, \ell_r}}(g \cdot (g^{-1} \cdot a_{\ell_1}) \otimes \cdots \otimes g \cdot (g^{-1} \cdot a_{\ell_r}) \otimes (g \cdot c)) \qquad (\iota_{k_{\ell_1, \ldots, \ell_r}} \ O(d)\text{-equivariant})$$

$$= \iota_{k_{\ell_1, \ldots, \ell_r}}(a_{\ell_1} \otimes \cdots \otimes a_{\ell_r} \otimes (g \cdot c)).$$

Hence, by taking the expectation with respect to the Haar probability measure of $O(d)$ and linearity of contractions, we have that

$$f(a_1, \ldots, a_n) = \iota_{k_{\ell_1, \ldots, \ell_r}} \left( a_{\ell_1} \otimes \cdots \otimes a_{\ell_r} \otimes \left( \mathop{\mathbb{E}}_{\mathfrak{g} \in O(d)} \mathfrak{g} \cdot c \right) \right), \tag{41}$$

where $\mathbb{E}_{\mathfrak{g} \in O(d)}$ is the expectation with respect the Haar probability measure of $O(d)$. This holds because

$$\begin{aligned}
f(a_1, \ldots, a_n) &= \mathop{\mathbb{E}}_{\mathfrak{g} \in O(d)} f(a_1, \ldots, a_n) \\
&= \mathop{\mathbb{E}}_{\mathfrak{g} \in O(d)} \iota_{k_{\ell_1, \ldots, \ell_r}} (a_{\ell_1} \otimes \cdots \otimes a_{\ell_r} \otimes (\mathfrak{g} \cdot c)) \\
&= \iota_{k_{\ell_1, \ldots, \ell_r}} \left( a_{\ell_1} \otimes \cdots \otimes a_{\ell_r} \otimes \left( \mathop{\mathbb{E}}_{\mathfrak{g} \in O(d)} \mathfrak{g} \cdot c \right) \right).
\end{aligned}$$

Now, $\mathbb{E}_{\mathfrak{g} \in O(d)} \mathfrak{g} \cdot c$ is an $O(d)$-isotropic tensor. Hence, we have shown that we can write $f$ in the desired form. $\qquad \square$

*Proof of Lemma 2.* Let $t \in \mathbb{R}$, since each $f_r$ is homogeneous of degree $r$, we have

$$f(t\, a_1, \ldots, t\, a_n) = \sum_{r=0}^{R} f_r(t\, a_1, \ldots, t\, a_n) = \sum_{r=1}^{R} t^r f_r(a_1, \ldots, a_n).$$

Let now $g \in O(d)$, then, by equivariance of $f$, we have

$$\sum_{r=0}^{R} t^r f_r(g \cdot a_1, \ldots, g \cdot a_n) = \sum_{r=0}^{R} t^r g \cdot f_r(a_1, \ldots, a_n), \tag{42}$$

since

$$\begin{aligned}
\sum_{r=0}^{R} t^r f_r(g \cdot a_1, \ldots, g \cdot a_n) &= f(t\,(g \cdot a_1), \ldots, t\,(g \cdot a_n)) \\
&= f(g \cdot t\, a_1, \ldots, g \cdot t\, a_n) \\
&= g \cdot f(t\, a_1, \ldots, t\, a_n) \\
&= g \cdot \sum_{r=0}^{R} t^r f_r(a_1, \ldots, a_n) \\
&= \sum_{r=0}^{R} t^r g \cdot f_r(a_1, \ldots, a_n).
\end{aligned}$$

Hence, for all $g \in O(d)$, $t \in \mathbb{R}$ and $(a_1, \ldots, a_n) \in \prod_{i=1}^{n} \mathcal{T}_{k_i}(\mathbb{R}^d, p_i)$, we have that

$$0 = \sum_{r=0}^{R} t^r \left( g \cdot f_r(a_1, \ldots, a_n) - f_r(g \cdot a_1, \ldots, g \cdot a_n) \right). \tag{43}$$

Now, the only way in which the univariate polynomial in $t$ of degree $R$ is identically zero is if it is the zero polynomial (cf. (Cox et al., 2015, Chapter 1 §1 Proposition 5)). Therefore for all $r \in \mathbb{N}$, $g \in O(d)$ and $(a_1, \ldots, a_n) \in \prod_{i=1}^{n} \mathcal{T}_{k_i}(\mathbb{R}^d, p_i)$,

$$f_r(g \cdot a_1, \ldots, g \cdot a_n) = g \cdot f_r(a_1, \ldots, a_n), \tag{44}$$

i.e., for each $r$, $f_r$ is $O(d)$-equivariant, as we wanted to show. $\qquad \square$

*Proof of Lemma 3.* First, we will show that if the decomposition exists, each summand is equivariant. Then, we will show that the decomposition exists.

Let $t_1, \ldots, t_n \in \mathbb{R}$. Then, by the linearity, we have that

$$f_r(t_1 \, a_1, \ldots, t_n \, a_n) = \sum_{1 \leq \ell_1 \leq \ldots \leq \ell_r \leq n} t_{\ell_1} \cdots t_{\ell_r} \, f_{\ell_1, \ldots, \ell_r}(a_{\ell_1}, \ldots, a_{\ell_r}), \tag{45}$$

since

$$f_r(t_1 \, a_1, \ldots, t_n \, a_n) = \sum_{1 \leq \ell_1 \leq \ldots \leq \ell_r \leq n} f_{\ell_1, \ldots, \ell_r}(t_{\ell_1} \, a_{\ell_1}, \ldots, t_{\ell_r} \, a_{\ell_r})$$

$$= \sum_{1 \leq \ell_1 \leq \ldots \leq \ell_r \leq n} t_{\ell_1} \cdots t_{\ell_r} \, f_{\ell_1, \ldots, \ell_r}(a_{\ell_1}, \ldots, a_{\ell_r}) \, .$$

Now, let $g \in O(d)$. Then, by the equivariance of $f_r$, we have

$$\sum_{1 \leq \ell_1 \leq \ldots \leq \ell_r \leq n} t_{\ell_1} \cdots t_{\ell_r} \, f_{\ell_1, \ldots, \ell_r}(g \cdot a_{\ell_1}, \ldots, g \cdot a_{\ell_r}) = \sum_{1 \leq \ell_1 \leq \ldots \leq \ell_r \leq n} t_{\ell_1} \cdots t_{\ell_r} \, g \cdot f_{\ell_1, \ldots, \ell_r}(a_{\ell_1}, \ldots, a_{\ell_r}),$$

$$\tag{46}$$

since

$$\sum_{1 \leq \ell_1 \leq \ldots \leq \ell_r \leq n} t_{\ell_1} \cdots t_{\ell_r} \, f_{\ell_1, \ldots, \ell_r}(g \cdot a_{\ell_1}, \ldots, g \cdot a_{\ell_r})$$

$$= f_r(t_1 \, (g \cdot a_1), \ldots, t_n \, (g \cdot a_n))$$

$$= f_r(g \cdot t_1 \, a_1, \ldots, g \cdot t_n \, a_n)$$

$$= g \cdot f_r(t_1 \, a_1, \ldots, t_n \, a_n)$$

$$= g \cdot \left( \sum_{1 \leq \ell_1 \leq \ldots \leq \ell_r \leq n} t_{\ell_1} \cdots t_{\ell_r} \, f_{\ell_1, \ldots, \ell_r}(a_{\ell_1}, \ldots, a_{\ell_r}) \right)$$

$$= \sum_{1 \leq \ell_1 \leq \ldots \leq \ell_r \leq n} t_{\ell_1} \cdots t_{\ell_r} \, g \cdot f_{\ell_1, \ldots, \ell_r}(a_{\ell_1}, \ldots, a_{\ell_r}).$$

Hence, for all $g \in O(d)$, $t \in \mathbb{R}$ and $(a_1, \ldots, a_n) \in \prod_{i=1}^n \mathcal{T}_{k_i}(\mathbb{R}^d, p_i)$, we have that

$$0 = \sum_{1 \leq \ell_1 \leq \ldots \leq \ell_r \leq n} t_{\ell_1} \cdots t_{\ell_r} \, [g \cdot f_{\ell_1, \ldots, \ell_r}(a_{\ell_1}, \ldots, a_{\ell_r}) - f_{\ell_1, \ldots, \ell_r}(g \cdot a_{\ell_1}, \ldots, g \cdot a_{\ell_r})]_{i_1, \ldots, i_{k'}}.$$

$$\tag{47}$$

Now, each of these is a polynomial in $t_1, \ldots, t_n$ that vanishes on $\mathbb{R}^n$. Moreover, note that no two $t_{\ell_1} \cdots t_{\ell_r}$ give the same monomial. Hence, by (Cox et al., 2015, Chapter 1 §1 Proposition 5), all these polynomials are the zero polynomial, i.e., their coefficients are zero. In this way, we conclude that for each $f_{\ell_1, \ldots, \ell_r}$, and all $g \in O(d)$, $t \in \mathbb{R}$ and $(a_1, \ldots, a_n) \in \prod_{i=1}^n \mathcal{T}_{k_i}(\mathbb{R}^d, p_i)$,

$$f_{\ell_1, \ldots, \ell_r}(g \cdot a_{\ell_1}, \ldots, g \cdot a_{\ell_r}) = g \cdot f_{\ell_1, \ldots, \ell_r}(a_{\ell_1}, \ldots, a_{\ell_r}), \tag{48}$$

i.e., each $f_{\ell_1, \ldots, \ell_r}$ is $O(d)$-equivariant.

Now, we show how to obtain the decomposition. Recall that $f_r$ is homogeneous of degree $r$. Therefore each entry of $f_r(a_1, \ldots, a_n)$ is an homogeneous polynomial of degree $r$ in the $[a_i]_{j_1, \ldots, j_{k_i}}$, i.e., a linear combination of products of the form

$$\prod_{q=1}^r [a_{\ell_q}]_{j_{q,1}, \ldots, j_{q, k_{\ell_q}}},$$

where, without loss of generality, we can assume that $\ell_1 \leq \cdots \leq \ell_q$. Hence, in coordinates, we have

$$[f_r(a_1, \ldots, a_n)]_{i_1, \ldots, i_{k'}}$$

$$= \sum_{1 \leq \ell_1 \leq \cdots \leq \ell_r \leq n} \lambda_{\ell_1, \ldots, \ell_r; i_1, \ldots, i_{k'}; j_{1,1}, \ldots, j_{1, k_{\ell_1}}, \ldots, j_{r,1}, \ldots, j_{r, k_{\ell_r}}} \prod_{q=1}^r [a_{\ell_q}]_{j_{q,1}, \ldots, j_{q, k_{\ell_q}}} \tag{49}$$

And so, we can consider the map $f_{\ell_1,\ldots,\ell_r}$ given in coordinates by

$$[f_{\ell_1,\ldots,\ell_r}(a_{\ell_1},\ldots,a_{\ell_r})]_{i_1,\ldots,i_{k'}} := \lambda_{\ell_1,\ldots,\ell_r;i_1,\ldots,i_{k'};j_{1,1},\ldots,j_{1,k_{\ell_1}},\ldots,j_{r,1},\ldots,j_{r,k_{\ell_r}}} \prod_{q=1}^{r}[a_{\ell_q}]_{j_{q,1},\ldots,j_{q,k_{\ell_q}}},$$

(50)

which, by construction, is the composition of the linear map given by

$$b \mapsto \lambda_{\ell_1,\ldots,\ell_r;i_1,\ldots,i_{k'};j_1,\ldots,j_{\sum_{q=1}^{r}k_{\ell_q}}}[b]_{j_1,\ldots,j_{\sum_{q=1}^{r}k_{\ell_q}}},$$

in coordinates, and $(a_{\ell_1},\ldots,a_{\ell_r}) \mapsto a_{\ell_1} \otimes \cdots \otimes a_{\ell_r}$. Hence the desired decomposition of $f_r$ has been obtained. $\qquad\square$

## C    PROOF OF COROLLARY 1

In this section, we will prove Corollary 1 using Lemma 1, which we prove afterward.

*Proof of Corollary 1.* By Theorem 1 and Lemma 1, we can assume, without loss of generality, it suffices to consider the special case where $f$ consists of a single term

$$f(v_1,\ldots,v_n) = \iota_r\left( v_{\ell_1} \otimes \cdots \otimes v_{\ell_r} \otimes \left(\delta^{\otimes \frac{r+k'}{2}}\right)^{\sigma}\right),$$

(51)

for some $\sigma \in S_{r+k'}$ and $r + k'$ even. To simplify notation, set $t := \frac{r+k'}{2}$.

Now, note that

$$\delta = e_i \otimes e_i,$$

where $\{e_1,\ldots,e_d\}$ is the canonical basis of $\mathbb{R}^d$. Hence we get

$$f(v_1,\ldots,v_n) = \iota_r(v_{\ell_1} \otimes \cdots \otimes v_{\ell_r} \otimes (e_{i_1} \otimes e_{i_1} \otimes \cdots \otimes e_{i_t} \otimes e_{i_t})^{\sigma})$$

(52)

Let's write

$$(e_{i_1} \otimes e_{i_1} \otimes \cdots \otimes e_{i_t} \otimes e_{i_t})^{\sigma} = e_{j_1} \otimes \cdots \otimes e_{j_{2t}}$$

where each $(j_1,\ldots,j_{2t})$ is some permutation of $(i_1,i_1,\ldots,i_t,i_t)$ and so, by Einstein notation, we are still adding over repeated indexes. Then we have that

$$f(v_1,\ldots,v_n) = \langle v_{\ell_1},e_{j_1}\rangle \cdots \langle v_{\ell_r},e_{j_r}\rangle e_{j_{r+1}} \otimes \cdots \otimes e_{j_{2t}}.$$

(53)

Now, we can freely rearrange the $\langle v_\ell,e_j\rangle$ as they are scalars. There are three cases for each of the original indices $i_q$: (a) $e_{i_q}$ appears in an inner product twice, (b) $e_{i_q}$ appears in an inner product once, or (c) $e_{i_q}$ does not appear in an inner product.

In the case (a), we will get

$$\langle v_\ell,e_i\rangle\langle v_{\ell'},e_i\rangle = \langle v_\ell,v_{\ell'}\rangle.$$

In the case (b), we will get

$$\langle v_\ell,e_i\rangle e_i = v_\ell.$$

And, in the case (c), we will get

$$e_i \otimes e_i = \delta.$$

Now, assume that we have $\alpha$ of the case (a), $\beta$ of the case (b) and $\gamma$ of the case (c). By permuting the $i_q$, which does not change the result, we can write for some permutation $\tilde{\sigma} \in S_{\beta+\gamma}$ and some permutation $J_1,\ldots,J_r$ some permutation of $\ell_1,\ldots,\ell_r$ that

$$f(v_1,\ldots,v_n) = \langle v_{J_1},e_{i_1}\rangle\langle v_{J_2},e_{i_1}\rangle \cdots \langle v_{J_{2\alpha-1}},e_{i_\alpha}\rangle\langle v_{J_{2\alpha}},e_{i_\alpha}\rangle$$

$$\left(\langle v_{J_{2\alpha+1}},e_{i_{\alpha+1}}\rangle e_{i_{\alpha+1}} \otimes \cdots \otimes \langle v_{J_{2\alpha+\beta}},e_{i_{\alpha+\beta}}\rangle e_{i_{\alpha+\beta}}\right.$$

$$\left.\otimes e_{i_{\alpha+\beta+1}} \otimes e_{i_{\alpha+\beta+1}} \otimes \cdots \otimes e_{i_{\alpha+\beta+\gamma}} \otimes e_{i_{\alpha+\beta+\gamma}}\right)^{\tilde{\sigma}}$$

$$= \langle v_{J_1},v_{J_2}\rangle \cdots \langle v_{J_{2\alpha-1}},v_{J_{2\alpha}}\rangle\left(v_{J_{2\alpha+1}} \otimes \cdots \otimes v_{2\alpha+\beta} \otimes \delta^{\otimes\gamma}\right)^{\tilde{\sigma}}.$$

Hence, the desired claim follows, and we finish the proof. $\qquad\square$

*Proof of Lemma 1.* We will prove each case separately. However, note that no matter the value of $p$, an $O(d)$-isotropic tensor is always an $SO(d)$-isotropic tensor since $\det(M(g)) = 1$ for all $g \in SO(d)$. Now, by (Jeffreys, 1973, Theorem §2) (cf. (Appleby et al., 1987, Eq. (4.10))), any $SO(d)$-isotropic tensor $z$ can be written as a linear combination of the form

$$z = \sum_{\sigma \in S_k} \alpha_\sigma \left( \delta^{\otimes \frac{k}{2}} \right)^\sigma + \beta_\sigma \left( \delta^{\otimes \frac{k-d}{2}} \otimes \epsilon \right)^\sigma, \tag{54}$$

where $\delta$ is the Kronecker delta (Definition 8), and $\epsilon$ is the Levi-Civita symbol (Definition 9), with the convention that the coefficients $\alpha_\sigma$ and $\beta_\sigma$ are zero when the expressions $\delta^{\otimes \frac{k}{2}}$ and $\delta^{\otimes \frac{k-d}{2}}$ do not make sense. More precisely, the $\alpha_\sigma = 0$ if $k$ is odd, and the $\beta_\sigma = 0$ if $k - d$ is odd.

Note that under the $SO(d)$-action, we don't need to worry about the parity, and so both $\delta$ and $\epsilon$ are $SO(d)$-invariant. However, for the $O(d)$-action, the parity matters. Suppose $\gamma \in O(d)$ is a hyperplane reflection, and let $T$ be an $O(d)$-isotropic $k_{(-)}$-tensor. If $\hat{T}$ is a $k_{(+)}$-tensor whose components equal $T$, then

$$\gamma \cdot \hat{T} = -\hat{T} \, .$$

Likewise, if $T$ is an $O(d)$-isotropic $k_{(+)}$-tensor and $\hat{T}$ is a $k_{(-)}$-tensor whose components equal $T$, then

$$\gamma \cdot \hat{T} = -\hat{T} \, .$$

Note that being isotropic depends on the parity because it affects the considered action.

*Case $p = -1$:* Let $z \in \mathcal{T}_k\left( \mathbb{R}^d, + \right)$ be $O(d)$-isotropic. In particular, $z$ is also $SO(d)$-isotropic, and so we can write it using (54).

Recall that $O(d)$ is generated by all the (hyperplane) reflections. Hence, to show that $z$ is an $O(d)$-isotropic, we need only to show that for every (hyperplane) reflection $\gamma \in O(d)$,

$$\gamma \cdot z = z.$$

Now, by our observation above, inside $\mathcal{T}_k\left( \mathbb{R}^d, + \right)$, we have that

$$\gamma \cdot \delta^{\otimes \frac{k-d}{2}} \otimes \epsilon = -\delta^{\otimes \frac{k-d}{2}} \otimes \epsilon, \tag{55}$$

since $\delta^{\otimes \frac{k-d}{2}} \otimes \epsilon$ is an $O(d)$-isotropic $k_{(-)}$-tensor. Hence

$$\begin{aligned} \gamma \cdot z &= \gamma \cdot \sum_{\sigma \in S_k} \alpha_\sigma \left( \delta^{\otimes \frac{k}{2}} \right)^\sigma + \beta_\sigma \left( \delta^{\otimes \frac{k-d}{2}} \otimes \epsilon \right)^\sigma \\ &= \sum_{\sigma \in S_k} \alpha_\sigma \left( (\gamma \cdot \delta)^{\otimes \frac{k}{2}} \right)^\sigma + \beta_\sigma \left( (\gamma \cdot \delta)^{\otimes \frac{k-d}{2}} \otimes \gamma \cdot \epsilon \right)^\sigma \\ &= \sum_{\sigma \in S_k} \alpha_\sigma \left( \delta^{\otimes \frac{k}{2}} \right)^\sigma - \beta_\sigma \left( \delta^{\otimes \frac{k-d}{2}} \otimes \epsilon \right)^\sigma \\ &= z - 2 \sum_{\sigma \in S_k} \beta_\sigma \left( \delta^{\otimes \frac{k-d}{2}} \otimes \epsilon \right)^\sigma \, . \end{aligned}$$

Now, by assumption, $z$ is $O(d)$-isotropic, so we can conclude $\sum_{\sigma \in S_k} \beta_\sigma \left( \delta^{\otimes \frac{k-d}{2}} \otimes \epsilon \right)^\sigma = 0$, and so $z$ has the desired form.

*Case $p = -1$:* We argue as above, but using that for a (hyperplane) reflection $\gamma \in O(d)$, we have, inside $\mathcal{T}_k\left( \mathbb{R}^d, - \right)$,

$$\gamma \cdot \delta^{\otimes \frac{k}{2}} = -\delta^{\otimes \frac{k}{2}}, \tag{56}$$

since $\delta^{\otimes \frac{k}{2}}$ is an $O(d)$-isotropic $k_{(+)}$-tensor. Hence, arguing similarly as in the previous case, we conclude that $\sum_{\sigma \in S_k} \alpha_\sigma \left( \delta^{\otimes \frac{k}{2}} \right)^\sigma = 0$, and so that $z$ has the desired form. $\qquad \square$

## D    SMALLER PARAMETERIZATION OF $O(d)$-ISOTROPIC TENSORS

In Lemma 1, the sum does not have to be over all permutations. The reason for this is that the tensors

$$\delta^{\otimes \frac{k}{2}} \quad \text{and} \quad \delta^{\otimes \frac{k-d}{2}} \otimes \epsilon$$

do not have a trivial stabilizer under the action of $S_k$. One can easily see the following proposition. Recall that the *stabilizer of a $k$-tensor $\pm T$ in $S_k$* is the following subgroup:

$$\mathrm{Stab}_{S_k}(\pm T) := \{\sigma \in S_k \mid T^\sigma = \pm T\}, \tag{57}$$

where $T^\sigma = \pm T$ means that either $T^\sigma = T$ or $T^\sigma = -T$. Note that the laxity in the signs comes from the fact that positive summands and their negative counterparts can be combined.

**Proposition 4.** (a) If $k$ is even, $\mathrm{Stab}_{S_k}\left(\pm\delta^{\otimes \frac{k}{2}}\right)$ is generated by the transpositions

$$(1,2),(3,4),\ldots,(k-1,k)$$

and all permutations of the form

$$(i,j)(i+1,j+1)$$

with $i,j < k$ odd. In particular, $\#\mathrm{Stab}_{S_k}\left(\pm\delta^{\otimes \frac{k}{2}}\right) = (k/2)!\,2^{k/2}$.

(b) If $k - d$ is even, $\mathrm{Stab}_{S_k}\left(\pm\delta^{\otimes \frac{k-d}{2}} \otimes \epsilon\right)$ is generated by the transpositions

$$(1,2),(3,4),\ldots,(k-d-1,k-d),$$

all permutations of the form

$$(i,j)(i+1,j+1)$$

with $i,j < k-d$ odd, and all transpositions of the form

$$(i,j)$$

with $k-d < i,j$. In particular, $\#\mathrm{Stab}_{S_k}\left(\pm\delta^{\otimes \frac{k-d}{2}} \otimes \epsilon\right) = ((k-d)/2)!\,2^{(k-d)/2}d!$.

*Proof.* This follows from (Roe Goodman, 2009, Theorem 5.3.4). $\qquad\square$

Using these proposition, we can write any $O(d)$-isotropic $k_{(+)}$-tensor as

$$\sum_{\sigma \in G_k} \alpha_\sigma \left(\delta^{\otimes \frac{k}{2}}\right)^\sigma$$

with the $\alpha_\sigma$ real and

$$G_k = \left\{\sigma \in S_k : \sigma(1) < \sigma(3) < \cdots < \sigma(k-1) \text{ and for all } i \leq \frac{k}{2},\, \sigma(2i-1) < \sigma(2i)\right\}$$

of size $\frac{k!}{(k/2)!2^{k/2}}$; and any $O(d)$-isotropic $k_{(-)}$-tensor as

$$\sum_{\sigma \in H_k} \beta_\sigma \left(\delta^{\otimes \frac{k-d}{2}} \otimes \epsilon\right)^\sigma \tag{58}$$

with the $\beta_\sigma$ real and

$$H_k = \Big\{\sigma \in S_k : \sigma(1) < \sigma(3) < \cdots < \sigma(k-d-1), \text{ for all } i \leq \frac{k-d}{2},\, \sigma(2i-1) < \sigma(2i)$$
$$\text{and for all } j > k-d, \sigma(j) < \sigma(j+1)\Big\}.$$

of size $\dfrac{k!}{\left(\frac{k-d}{2}\right)!2^{\frac{k-d}{2}}d!}$.

## E    EXAMPLE OF THEOREM 1

In this section, we give a second example of Theorem 1.

**Example 2.** Let $f : \mathcal{T}_1(\mathbb{R}^d, +) \times \mathcal{T}_2(\mathbb{R}^d, +) \to \mathcal{T}_2(\mathbb{R}^d, +)$ be $O(d)$-equivariant polynomial of degree at most 2. By Theorem 1 we can write $f$ in the form

$$f(a_1, a_2) = \sum_{r=0}^{2} \sum_{1 \leq \ell_1 \leq \cdots \leq \ell_r \leq 2} \iota_{k_{\ell_1, \ldots, \ell_r}} (a_{\ell_1} \otimes \ldots \otimes a_{\ell_r} \otimes c_{\ell_1, \ldots, \ell_r}) , \tag{59}$$

where $c_{\ell_1, \ldots, \ell_r}$ is an $O(d)$-isotropic $(k_{\ell_1, \ldots, \ell_r} + 2)_{(+)}$-tensor. By Lemma 1, $c_{\ell_1, \ldots, \ell_r}$ is nontrivial only when $k_{\ell_1, \ldots, \ell_r} + 2$ is even. Recall that $k_{\ell_1, \ldots, \ell_r} = \sum_{q=1}^{r} k_{\ell_q}$. The inputs are a $1_{(+)}$-tensor and $2_{(+)}$-tensor. The even combinations of 1 and 2 with at most 2 terms are $\emptyset, 2, 1+1, 2+2$ so we have

$$f(a_1, a_2) = \beta_0 \delta + \iota_2(a_2 \otimes c_2) + \iota_2(a_1 \otimes a_1 \otimes c_2') + \iota_4(a_2 \otimes a_2 \otimes c_3) , \tag{60}$$

where $c_2, c_2'$ are $O(d)$-isotropic $4_{(+)}$-tensors and $c_3$ is an $O(d)$-isotropic $6_{(+)}$-tensor. By similar reasoning to Example 1, we can write

$$\iota_2(a_2 \otimes c_2) = \beta_1 \operatorname{tr}(a_2)\delta + \beta_2 a_2 + \beta_3 a_2^\top \tag{61}$$

for constants $\beta_1, \beta_2, \beta_3$ and

$$\iota_2(a_1 \otimes a_1 \otimes c_2') = \beta_4 \langle a_1, a_1 \rangle \delta + \beta_5 a_1 \otimes a_1 , \tag{62}$$

for constants $\beta_4, \beta_5$ (there are only two terms due to the symmetry of $a_1 \otimes a_1$). It remains to consider $\iota_4(a_2 \otimes a_2 \otimes c_3)$. By Lemma 1, we can write

$$c_3 = \sum_{\sigma \in G_6} \beta_\sigma (\delta^{\otimes 3})^\sigma , \tag{63}$$

where $|G_6| = 6!/(3!2^3) = 15$. In particular, we have

$$G_6 = \big\{ (1,2,3,4,5,6), (1,2,3,5,4,6), (1,2,3,5,6,4), (1,3,2,4,5,6), (1,3,2,5,4,6),$$
$$(1,3,2,5,6,4), (1,3,4,2,5,6), (1,3,4,5,2,6), (1,3,4,5,6,2), (1,3,5,2,4,6),$$
$$(1,3,5,2,6,4), (1,3,5,4,2,6), (1,3,5,4,6,2), (1,3,5,6,2,4), (1,3,5,6,4,2) \big\} .$$

However, due to the symmetry of $a_2 \otimes a_2$, when we compute $\iota_4\big(a_2 \otimes a_2(\delta^{\otimes 3})^\sigma\big)$ for $\sigma \in G_6$, there are only 7 distinct terms

$$\iota_4(a_2 \otimes a_2 \otimes c_3) =$$
$$\beta_6 \operatorname{tr}(a_2)^2 \delta + \beta_7 \operatorname{tr}(a_2)a_2 + \beta_8 \operatorname{tr}(a_2)a_2^\top + \beta_9 a_2^\top a_2 + \beta_{10} a_2 a_2^\top + \beta_{11} a_2 a_2 + \beta_{12} a_2^\top a_2^\top . \tag{64}$$

In summary,

$$f(a_1, a_2) = \beta_0 \delta + \beta_1 \operatorname{tr}(a_2)\delta + \beta_2 a_2 + \beta_3 a_2^\top + \beta_3 \langle a_1, a_1 \rangle \delta + \beta_4 a_1 \otimes a_1 + \beta_5 \operatorname{tr}(a_2)^2 \delta$$
$$+ \beta_6 \operatorname{tr}(a_2)a_2 + \beta_7 \operatorname{tr}(a_2)a_2^\top + \beta_8 a_2^\top a_2 + \beta_9 a_2 a_2^\top + \beta_{10} a_2 a_2 + \beta_{11} a_2^\top a_2^\top ,$$

for some coefficients $\beta_0, \beta_1, \ldots, \beta_{11}$.

## F    GENERALIZATION TO OTHER LINEAR ALGEBRAIC GROUPS

In this section, we will show how Theorem 1 and Corollary 1 can be extended to the indefinite orthogonal and the symplectic group as Theorem 2 and Corollary 2.

The main idea to extend Theorem 1 to other groups is to use some form of averaging. On $O(d)$, the compactness guarantees the existence of a Haar probability measure. However, to apply the same trick over non-compact groups such as $O(s, d-s)$ and $Sp(d)$, we need to use technical machinery to imitate the averaging strategy.

First, we introduce some definitions and examples regarding complex and real linear algebraic groups. The main point will be to establish how to get a compact subgroup over which to average. Basically the results will generalize to real linear algebraic groups such that their complexifications have a Zariski-dense compact subgroup. For instance reductive connected complex algebraic groups satisfy this assumption. Second, we prove a generalization of Theorem 1 for complex linear algebraic groups with a Zariski-dense compact subgroup acting on complex tensors. Third, we prove a generalization of Theorem 1 for real linear algebraic groups that are compact or such that their complexification has a Zariski-dense compact subgroup. Finally, we prove Corollary 2.

## F.1 REDUCTIVE COMPLEX AND REAL LINEAR ALGEBRAIC GROUPS

Recall that a *complex linear algebraic group* is a subgroup $G$ of $GL(V)$, where $V$ is a finite-dimensional complex vector space, such that $G$ is the zero set of some set of complex polynomial functions over $End(V)$, the set of (complex) linear maps $V \to V$. Recall also that a *rational G-module* of $G$ is a vector space $U$ together with a linear action of $G$ on $U$ such that the map $G \times U \ni (g,x) \mapsto g \cdot x \in U$ is polynomial[2], and that a $G$-submodule $U_0$ of $U$ is a vector subspace $U_0 \subseteq U$ such that for all $g \in G$, $g \cdot U_0 \subseteq U_0$.

**Definition 10.** (Roe Goodman, 2009, Def. 3.3.1) A *reductive complex linear algebraic group* is a complex linear algebraic group $G \subset GL(V)$ such that every rational $G$-module $U$ is *completely reducible*, i.e., for every $G$-submodule $U_0$ of $U$, there is a $G$-submodule $U_1$ such that $U = U_0 + U_1$ and $U_0 \cap U_1 = 0$.

**Example 3.** Given any finite-dimensional vector space, the classical complex groups $GL(V)$ and $SL(V)$ are reductive complex linear algebraic groups.

**Example 4.** Given any finite-dimensional vector space $V$ together with a symmetric non-degenerate bilinear form[3] $\langle \cdot, \cdot \rangle : V \times V \to \mathbb{C}$, the (complex) orthogonal group

$$O(V, \langle \cdot, \cdot \rangle) := \{g \in GL(V) \mid \text{for all } v, w \in V, \langle g \cdot v, g \cdot w \rangle = \langle v, w \rangle\} \tag{65}$$

is a reductive complex linear algebraic group. We will pay special attention to the following family of complex orthogonal groups:

$$O^{\mathbb{C}}(s, d-s) := \{g \in GL(\mathbb{C}^d) \mid g^\top \mathbb{I}_{s,d-s} g = \mathbb{I}_{s,d-s}\} = O(\mathbb{C}^d, \langle \cdot, \cdot \rangle_s) \tag{66}$$

where $\langle u, v \rangle_s := u^\top \mathbb{I}_{s,d-s} v$. Note that all these groups are isomorphic, satisfying that

$$O^{\mathbb{C}}(s, d-s) = \begin{pmatrix} \mathbb{I}_s & \\ & i\mathbb{I}_{d-s} \end{pmatrix} O^{\mathbb{C}}(d, 0) \begin{pmatrix} \mathbb{I}_s & \\ & i\mathbb{I}_{d-s} \end{pmatrix}^{-1}.$$

Moreover, this is true in general: any two complex orthogonal groups are isomorphic if they are of the same order—this follows from the fact that all symmetric non-degenerate bilinear forms are equivalent over the complex numbers.

**Example 5.** Given any finite-dimensional vector space $V$ together with an anti-symmetric non-degenerate bilinear form $\langle \cdot, \cdot \rangle : V \times V \to \mathbb{C}$, the (complex) symplectic group

$$Sym(V, \langle \cdot, \cdot \rangle) := \{g \in GL(V) \mid \text{for all } v, w \in V, \langle g \cdot v, g \cdot w \rangle = \langle v, w \rangle\} \tag{67}$$

is a reductive complex linear algebraic group. We will pay special attention to the following special case:

$$Sp^{\mathbb{C}}(d) := \{g \in GL(\mathbb{C}^d) \mid g^\top J_d g = J_d\} = Sp(\mathbb{C}^d, \langle \cdot, \cdot \rangle_{\text{symp}}) \tag{68}$$

where $\langle u, v \rangle_{\text{symp}} := u^\top J_d v$. Note that any symplectic group of order $d$ is isomorphic to $Sp^{\mathbb{C}}(d)$ because any two antisymmetric non-degenerate bilinear forms are equivalent over the complex numbers.

**Example 6.** The complex linear algebraic group

$$H = \left\{ \begin{pmatrix} 1 & t \\ & 1 \end{pmatrix} \mid t \in \mathbb{C} \right\}$$

is not reductive, since $\mathbb{C}^2$ is an $H$-module that is not completely reducible. Note that $\mathbb{C} \times 0$ is the only $H$-submodule of $\mathbb{C}^2$, so we cannot find a complementary $H$-submodule.

Recall that a subset $X$ of a set $\tilde{X}$ is *Zariski-dense* in $\tilde{X}$ if every polynomial function that vanishes in $X$ vanishes in $\tilde{X}$, i.e. if every polynomial function that does not vanish on $\tilde{X}$ does not vanish in $X$. The following theorem allows us to use the power of averaging for reductive connected complex linear algebraic groups.

---

[2]To be precise, we mean that the map $G \times U \to U$ is a morphism of algebraic varieties. Choose basis for $U$ and $V$, so that we can identify $V$ with $\mathbb{C}^d$ and $U$ with $\mathbb{C}^n$. Then, being a morphism between algebraic varieties, just means that the map $G \times \mathbb{C}^n \to \mathbb{C}^n$ is the restriction of a map $\mathbb{C}^{d \times d} \times \mathbb{C}^n \to \mathbb{C}^n$ that can be written componentwise as $(p_l((g_{i,j})_{i,j}, (u_k)_k)/(\det g)^{a_l})_l$ where each $p_l$ is a polynomial in the $g_{i,j}$ and $u_k$ and each $a_l$ an integer.

[3]Recall that this means that for all $u, v, w \in V$ and $t, s \in \mathbb{C}$: (a) $\langle u, v \rangle = \langle v, u \rangle$, (b) for all $x \in V$, $\langle u, x \rangle = 0$ if and only if $u = 0$, and (c) $\langle tu + sv, w \rangle = t\langle u, w \rangle + s\langle v, w \rangle$.

**Theorem 3.** (Roe Goodman, 2009, Theorem 11.5.1) Let $G$ be a reductive connected complex algebraic group. Then there exists a Zariski-dense compact subgroup $K$. More precisely, there is a subgroup $U(G)$ of $G$ that is Zariski-dense in $G$ and that, with respect to the usual topology[4], is compact.

**Remark 5.** Note that using this compact subgroup $K$, we can consider expressions of the form

$$\underset{\mathfrak{u} \in U(G)}{\mathbb{E}} \mathfrak{u} \cdot T$$

by taking the expectation with respect to the unique Haar probability measure of $K$. Now, since $U(G)$ is Zariski-dense in $G$, we have that the fact that for all $u \in U(G)$, $u \cdot \left(\mathbb{E}_{\mathfrak{u} \in U(G)} \mathfrak{u} \cdot T\right) = \mathbb{E}_{\mathfrak{u} \in U(G)} \mathfrak{u} \cdot T$ implies that for all $g \in G$,

$$g \cdot \left(\underset{\mathfrak{u} \in U(G)}{\mathbb{E}} \mathfrak{u} \cdot T\right) = \underset{\mathfrak{u} \in U(G)}{\mathbb{E}} \mathfrak{u} \cdot T.$$

Note that $U(G)$ is not necessarily unique.

**Example 7.** In $GL(\mathbb{C}^d)$, the Zariski-dense compact subgroup is the group of unitary matrices:

$$U(\mathbb{C}^d) := \{g \in GL(\mathbb{C}^d) \mid g^* g = \mathbb{I}_d\}$$

where $^*$ denotes the conjugate transpose. In $SL(\mathbb{C}^d)$, it is the group of special unitary transformations:

$$SU(\mathbb{C}^d) := \{g \in U(\mathbb{C}^d) \mid \det g = 1\}.$$

**Example 8.** In $O^{\mathbb{C}}(s, d - s)$, the Zariski-dense compact subgroup is

$$\begin{pmatrix} \mathbb{I}_s & \\ & i\mathbb{I}_{d-s} \end{pmatrix} O(d) \begin{pmatrix} \mathbb{I}_s & \\ & i\mathbb{I}_{d-s} \end{pmatrix}^{-1}.$$

Note that when $s = 0$ or $s = d$, this is the orthogonal group over the reals. Moreover, this does not follow from Theorem 3 as $O^{\mathbb{C}}(s, d - s)$ is not connected.

**Example 9.** In $Sp^{\mathbb{C}}(d)$, the Zariski-dense compact subgroup is the so-called compact symplectic group:

$$USp(d) := Sp^{\mathbb{C}}(d) \cap U(\mathbb{C}^d).$$

Recall that a *real linear algebraic group* is a subgroup $G$ of $GL(V)$, where $V$ is a finite-dimensional real vector space, such that $G$ is the zero set of some set of real polynomial functions over $\mathbb{R}^{d \times d}$. Similarly, as we did with complex linear algebraic groups, we can talk about *rational modules* and about *reductive real linear algebraic groups*.

However, given a reductive real linear algebraic group we cannot necessarily guarantee the existence of a Zariski-dense compact subgroup. This means that we cannot apply the averaging trick directly, but we can do so by passing to the Zariski-dense compact subgroup of the complexification of the real linear algebraic group.

**Definition 11.** Let $G \subset GL(V)$ be a real linear algebraic group. The *complexification* $G^{\mathbb{C}}$ of $G$ is the complex linear algebraic group given by

$$G^{\mathbb{C}} := \{g \in GL(V^{\mathbb{C}}) \mid \text{for every polynomial } f \text{ such that } f(G) = 0, \ f(g) = 0\} \qquad (69)$$

where $V^{\mathbb{C}} := V \otimes_{\mathbb{R}} \mathbb{C}$ is the complexification of $V$, i.e., the complex vector space obtained from $V$ by extending scalars.

**Remark 6.** In essence, we complexify the underlying real algebraic variety. Group multiplication preserves its structure as a complex variety as it is given by polynomial functions of the matrix entries.

**Definition 12.** A real linear algebraic group $G$ is *complexly averageable* if it's Zariski-dense in its complexification and its complexification admits a Zariski-dense compact subgroup closed under complex conjugation.

**Remark 7.** Recall that the complexification of $\mathbb{R}^d$ is naturally isomorphic to $\mathbb{C}^d$.

**Example 10.** The complexification of $GL(\mathbb{R}^d)$ is $GL(\mathbb{C}^d)$, and the complexification of $SL(\mathbb{R}^d)$ is $SL(\mathbb{R}^d)$.

---

[4]The topology inherited from the Euclidean topology of $GL(\mathbb{C}^d)$.

**Example 11.** We have that
$$O(s, d-s)^{\mathbb{C}} = O^{\mathbb{C}}(s, d-s)$$
and that
$$Sp(d)^{\mathbb{C}} = Sp^{\mathbb{C}}(d).$$
Hence, both the indefinite orthogonal group and symplectic group are complexly averageable. The symplectic group is connected but the indefinite orthogonal group is not connected. However, it does have a Zariski-dense compact subgroup (see Example 8).

The following proposition shows that complexly averageable real linear algebraic groups are common.

**Proposition 5.** Let $G \subset GL(V)$ be a real linear algebraic group. (1) $G$ is Zariski-dense in $G^{\mathbb{C}}$. (2) If the complexification $G^{\mathbb{C}}$ of $G$ is connected and reductive, then $G$ is complexly averageable.

*Proof.* (1) Let $f$ be a complex polynomial vanishing on $G$. Then, we can write this polynomial as $f = f_r + if_i$ for some polynomials $f_r$ and $f_i$ with real coefficients. Now, since $f$ vanishes on $G$, then $f_r$ and $f_i$ vanish also on $G$—as otherwise there would be $g \in G$ such that either $f_r(g) \neq 0$ or $f_i(g) \neq 0$, contradicting $f(g) = 0$. But then, by definition of $G^{\mathbb{C}}$, $f_r$ and $f_i$ vanish on $G$ and so $f = f_r + if_i$ vanishes on $G^{\mathbb{C}}$. Hence we have just proven that a complex polynomial vanishes on $G$ if and only if vanishes on $G^{\mathbb{C}}$, i.e., we have proven that $G$ is Zariski-dense in $G^{\mathbb{C}}$.

(2) This follows from Theorem 3. $\qquad\square$

**Example 12.** Observe that the Zariski-dense compact subgroups of $O^{\mathbb{C}}(s, d-s)$ and $Sp^{\mathbb{C}}(d)$ that have been given satisfy that they are closed under the complex conjugation.

### F.2 COMPLEX EQUIVARIANT TENSOR MAPS

We will consider vector spaces on which a non-degenerate bilinear form has been chosen.

**Definition 13.** A *self-paired vector space* $(V, \langle \cdot, \cdot \rangle)$ is a finite-dimensional vector space $V$ together with a non-degenerate bilinear form $\langle \cdot, \cdot \rangle : V \times V \to \mathbb{C}$.

Recall the *universal property* of tensor products of vector spaces, by which multilinear maps $V_1 \times \cdots \times V_k \to W$ can be lifted to linear maps $V_1 \otimes \cdots \otimes V_k \to W$. Using the universal property, we can see that from a self-paired vector space $(V, \langle \cdot, \cdot \rangle)$, we get the family
$$(V^{\otimes k}, \langle \cdot, \cdot \rangle)$$
of self-paired spaces of tensors, by extending by linearity the expression
$$\langle v_1 \otimes \cdots \otimes v_k, \tilde{v}_1 \otimes \cdots \otimes \tilde{v}_k \rangle = \langle v_1, \tilde{v}_1 \rangle \cdots \langle v_k, \tilde{v}_k \rangle. \tag{70}$$
And again, by the universal property, we get a $k$-*contraction*
$$\iota_k : V^{\otimes(2k+k')} \cong V^{\otimes k} \otimes V^{\otimes k} \otimes V^{k'} \to V^{k'} \tag{71}$$
by extending by linearity, the expression
$$a \otimes b \otimes c \mapsto \langle a, b \rangle c. \tag{72}$$

Now, in the above setting, let $G$ be a group acting in a structure-preserving way on $(V, \langle \cdot, \cdot \rangle)$, meaning that the action is linear and preserves $\langle \cdot, \cdot \rangle$, i.e., for all $g \in G$, $v, \tilde{v} \in V$, $\langle v, \tilde{v} \rangle = \langle g \cdot v, g \cdot \tilde{v} \rangle$. Then, by the universal property, we get that $G$ acts also on $(V^{\otimes k}, \langle \cdot, \cdot \rangle)$ by extending linearly the expression
$$g(v_1 \otimes \cdots \otimes v_k) = \chi(g)(gv_1) \otimes \cdots \otimes (gv_k). \tag{73}$$
Moreover, by considering all (rational)[5] unidimensional representations $\chi : G \to \mathbb{C}^*$ of $G$, we get the following family of self-paired (rational) $G$-modules:
$$\mathcal{T}_k(V, \chi) := (V^{\otimes k}, \langle \cdot, \cdot \rangle) \tag{74}$$
where the action by $G$ is given by
$$g \cdot T := \chi(g) M(g) \cdot T \tag{75}$$

---

[5]Recall that rational means that the homomorphism is given by polynomials.

in a structure-preserving way. For the sake of distinction, we will denote the $k$-contraction as

$$\iota_k^G : \mathcal{T}_{2k+k'}(V, \chi) \to \mathcal{T}_{k'}(V, \chi) \tag{76}$$

in this setting to emphasize the dependence on the group $G$, as we will be choosing the original $\langle \cdot, \cdot \rangle$ in terms of the group. Using the universal property, we can easily see the following:

**Proposition 6.** The following statements hold:

    (a) The outer product map

$$\mathcal{T}_k(V, \chi) \times \mathcal{T}_{k'}(V, \chi') \to \mathcal{T}_{k+k'}(V, \chi\chi')$$

        is a $G$-equivariant bilinear map.

    (b) The $k$-contraction $\iota_k^G : \mathcal{T}_{2k+k'}(V, \chi) \to \mathcal{T}_{k'}(V, \chi)$ is a $G$-equivariant linear map.

    (c) For any $\sigma \in S_k$, the tensor index permutation by $\sigma$, $\mathcal{T}_k(V, \chi) \to \mathcal{T}_k(V, \chi)$ given by $v_1 \otimes \cdots \otimes v_k \mapsto v_{\sigma^{-1}(1)} \otimes \cdots \otimes v_{\sigma^{-1}(k)}$, is a $G$-equivariant linear map.

Finally, recall that a $G$-isotropic tensor of $\mathcal{T}_\ell(V, \chi)$ is a $G$-invariant tensor in $\mathcal{T}_\ell(V, \chi)$. Further, recall that an *entire* function is an analytic function whose Taylor series at any point has an infinite radius of convergence. We can now state the theorem.

**Theorem 4.** Let $G \subset GL(V)$ be a reductive connected complex linear algebraic group (or more generally, a complex linear algebraic group with a Zariski-dense compact subgroup) acting rationally on an structure-preserving way on a self-paired complex vector space $(V, \langle \cdot, \cdot \rangle)$ and $f : \prod_{i=1}^n \mathcal{T}_{k_i}(V, \chi_i) \to \mathcal{T}_{k'}(V, \chi')$ a $G$-equivariant entire function. Then we may write $f$ as follows:

$$f(a_1, \ldots, a_n) = \sum_{r=0}^\infty \sum_{1 \le \ell_1 \le \cdots \le \ell_r \le n} \iota_{k_{\ell_1, \ldots, \ell_r}}^G (a_{\ell_1} \otimes \ldots \otimes a_{\ell_r} \otimes c_{\ell_1, \ldots, \ell_r}) \tag{77}$$

where $c_{\ell_1, \ldots, \ell_r} \in \mathcal{T}_{k_{\ell_1, \ldots, \ell_r} + k'}(\mathbb{R}^d, \chi_{\ell_1, \ldots, \ell_r} \chi')$ is a $G$-isotropic tensor for $k_{\ell_1, \ldots, \ell_r} := \sum_{q=1}^r k_{\ell_q}$ and $\chi_{\ell_1, \ldots, \ell_r} = \prod_{q=1}^r \chi_{\ell_q}$.

To prove this, we proceed as in the orthogonal case: we reduce to the multihomogeneous case and then prove the result using averaging over the Zariski-dense compact subgroup.

**Lemma 4.** Let $G \in GL(V)$ be any subgroup acting linearly on a self-paired complex vector space $(V, \langle \cdot, \cdot \rangle)$ and $f : \prod_{i=1}^n \mathcal{T}_{k_i}(V, \chi_i) \to \mathcal{T}_{k'}(V, \chi')$ an entire function. Then, we can write $f$ as

$$f_r(a_1, \ldots, a_n) = \sum_{r=0}^\infty \sum_{1 \le \ell_1 \le \ldots \le \ell_r \le n} f_{\ell_1, \ldots, \ell_r}(a_{\ell_1}, \ldots, a_{\ell_r}), \tag{78}$$

where $f_{\ell_1, \ldots, \ell_r} : \prod_{i=1}^r \mathcal{T}_{k_{\ell_i}}(V, \chi_i) \to \mathcal{T}_{k'}(V, \chi')$ is the composition of the map

$$\prod_{i=1}^r \mathcal{T}_{k_{\ell_i}}(V, \chi_{\ell_i}) \to \mathcal{T}_{\sum_{i=1}^r k_{\ell_i}} \left( V, \prod_{i=1}^r \chi_{\ell_i} \right)$$

$$(a_{\ell_1}, \ldots, a_{\ell_r}) \mapsto a_{\ell_1} \otimes \ldots \otimes a_{\ell_r}$$

with a linear map $\mathcal{T}_{\sum_{i=1}^r k_{\ell_i}}(V, \prod_{i=1}^r \chi_{\ell_i}) \to \mathcal{T}_{k'}(\mathbb{R}^d, \chi')$.

Moreover, for the above decomposition, if $f$ is $G$-equivariant, then so are the $f_{\ell_1, \ldots, \ell_r}$.

**Remark 8.** Note that we don't need to assume anything about $G$ in the above lemma.

*Proof of Theorem 4.* By Lemma 4, we can assume without loss of generality that $f$ is of the form

$$f(a_1, \ldots, a_n) = \lambda(a_{\ell_1} \otimes \cdots \otimes a_{\ell_r})$$

for some non-negative integer $r$, $1 \le \ell_1 \le \cdot \le \ell_r \le r$ and $\lambda : \mathcal{T}_{\sum_{i=1}^r k_{\ell_i}}(V, \prod_{i=1}^r \chi_{\ell_i}) \to \mathcal{T}_{k'}(V, \chi')$ is linear.

The above map can be written as a linear combination of maps of the form

$$(a_1, \ldots, a_n) \mapsto \left( \prod_{i=1}^{r} \lambda_i(a_{\ell_i}) \right) v_{j_1} \otimes \cdots \otimes v_{j_{k'}}$$

where the $\lambda_{i,j}$ are linear and $v_j \in V$, due to the universal property—the factor $(\prod_{i=1}^{r} \lambda_{i,j}(a_{\ell_i}))$ just corresponds to a linear map $\mathcal{T}_{\sum_{i=1}^{r} k_{\ell_i}}(V, \prod_{i=1}^{r} \chi_{\ell_i}) \to \mathbb{C}$. Moreover,

$$\left( \prod_{i=1}^{r} \lambda_i(a_{\ell_i}) \right) v_{j_1} \otimes \cdots \otimes v_{j_{k'}} = \iota_{\sum_{i=1}^{r} k_{\ell_i} + k'}^{G} \left( a_{\ell_1} \otimes \cdots \otimes a_{\ell_r} \otimes c_1 \otimes \cdots \otimes c_r \otimes v_{j_1} \otimes \cdots \otimes v_{j_{k'}} \right)$$

where the $c_i \in \mathcal{T}_{k_i}(V, \chi_i)$ are the unique tensors so that for all $a_{\ell_i} \in \mathcal{T}_{k_i}(V, \chi_i)$,

$$\lambda_i(a_{\ell_i}) = \langle a_{\ell_i}, c_i \rangle.$$

These $c_i$ exist, because $\langle \cdot, \cdot \rangle$ is non-degenerate. Hence for some $c \in \mathcal{T}_{\sum_{i=1}^{r} k_{\ell_i} + k'}(V, \prod_{i=1}^{r} \chi_i \chi')$, we have

$$f(a_1, \ldots, a_n) = \iota_{\sum_{i=1}^{r} k_{\ell_i} + k'}^{G}(a_{\ell_1} \otimes \cdots \otimes a_{\ell_r} \otimes c). \tag{79}$$

Since $f$ and $\iota_{k_\ell + k'}(\cdot)$ are $G$-equivariant, we have that for all $g \in G$ and $a \in \prod_{i=1}^{n} \mathcal{T}_{k_i}(V, \chi_i)$,

$$\begin{aligned}
\iota_{k_\ell + k'}(a_{\ell_1} \otimes \cdots \otimes a_{\ell_r} \otimes c_\ell) &= f(a_1, \ldots, a_n) \\
&= f(g \cdot (g^{-1} \cdot a_1), \ldots, g \cdot (g^{-1} \cdot a_n)) \\
&= g \cdot f((g^{-1} \cdot a_1), \ldots, (g^{-1} \cdot a_n)) \\
&= g \cdot \iota_{k_\ell + k'} \left( (g^{-1} \cdot a_{\ell_1}) \otimes \cdots \otimes (g^{-1} \cdot a_{\ell_r}) \otimes c_\ell \right) \\
&= \iota_{k_\ell + k'}(a_{\ell_1} \otimes \cdots \otimes a_{\ell_r} \otimes (g \cdot c_\ell)).
\end{aligned}$$

Finally, $G$ has a Zariski-dense compact subgroup $U(G)$. Hence, averaging over $U(G)$, we can substitute $c$ by the $U(G)$-isotropic tensor

$$\mathbb{E}_{\mathfrak{u} \in U(G)} \mathfrak{u} \cdot c$$

where the expectation is taken with respect the unique Haar probability measure of $U(G)$. But, since $U(G)$ is Zariski-dense in $G$ and the action rational, $\mathbb{E}_{\mathfrak{u} \in U(G)} \mathfrak{u} \cdot c$ is also $G$-isotropic, as we wanted to show. $\qquad \square$

*Proof of Lemma 4.* Recall that, since $f$ in entire, we have, by Taylor's theorem, that

$$f(a) = \sum_{r=0}^{\infty} \frac{1}{r!} D_0^r f(a, \ldots, a) \tag{80}$$

where $a = (a_1, \ldots, a_n) \in \prod_{i=1}^{n} \mathcal{T}_{k_i}(V, \chi_i)$ and $D_0^k f : \left( \prod_{i=1}^{n} \mathcal{T}_{k_i}(V, \chi_i) \right)^k \to \mathcal{T}_{k'}(V, \chi')$ is the $k$-multilinear map given by $k$th order partial derivatives of $f$ at $0$. Now, write $a = a_1 + \cdots + a_n$ as an abuse of notation for

$$a = (a_1, 0, \ldots, 0) + \cdots + (0, \ldots, 0, a_n).$$

We will further use this abuse of notation to write $a_i$ instead of $(0, \ldots, 0, a_i, 0, \ldots, 0)$. Now, since $D_0^k f$ is $k$-multilinear and symmetric, we have that

$$\frac{1}{r!} D_0^r f(a, \ldots, a) = \sum_{1 \le \ell_1 \le \cdots \le \ell_r \le n} \frac{1}{\alpha_{\ell_1, \ldots, \ell_r}!} D_0^r f(a_{\ell_1}, \ldots, a_{\ell_r}) \tag{81}$$

where $\alpha_{\ell_1, \ldots, \ell_r} \in \mathbb{N}^r$ is the vector given by $(\alpha_{\ell_1, \ldots, \ell_r})_i := \#\{j \mid \ell_j = i\}$ and $\alpha! := \alpha_1! \cdots \alpha_r!$. Note that this terms appears when we reorder $(a_{\ell_1}, \ldots, a_{\ell_r})$ so that the subindices are in order.

Summing up, we can write $f$ as (78), with

$$f_{\ell_1, \ldots, \ell_r}(a_1, \ldots, a_n) = \frac{1}{\alpha_{\ell_1, \ldots, \ell_r}!} D_0^r f(a_{\ell_1}, \ldots, a_{\ell_r}), \tag{82}$$

where this has the desired form by the universal property of tensor products. Now, observe that for $t_1, \ldots, t_n \in \mathbb{C}$ and $(a_1, \ldots, a_n) \in \prod_{i=1}^n \mathcal{T}_{k_i}(V, \chi_i)$,

$$f_{\ell_1, \ldots, \ell_r}(t_1 a_1, \ldots, t_n a_n) = t^{\alpha_{\ell_1, \ldots, \ell_r}} f_{\ell_1, \ldots, \ell_r}(a_1, \ldots, a_n) \tag{83}$$

where $t^{\alpha_{\ell_1, \ldots, \ell_r}} := t_1^{\alpha_1} \cdots t_n^{\alpha_n}$. Hence, arguing as in Lemma 3, we have that for any $g \in G$ and all $(a_1, \ldots, a_n) \in \prod_{i=1}^n \mathcal{T}_{k_i}(V, \chi_i)$,

$$\sum_{r=0}^{\infty} \sum_{1 \leq \ell_1 \leq \cdots \leq \ell_r \leq n} t^{\alpha_{\ell_1, \ldots, \ell_r}} g \cdot f_{\ell_1, \ldots, \ell_r}(a_1, \ldots, a_n)$$

$$= \sum_{r=0}^{\infty} \sum_{1 \leq \ell_1 \leq \cdots \leq \ell_r \leq n} t^{\alpha_{\ell_1, \ldots, \ell_r}} f_{\ell_1, \ldots, \ell_r}(g \cdot a_1, \ldots, g \cdot a_n). \tag{84}$$

Hence, by the uniqueness of coefficients for entire functions functions that are equal[6], we conclude that for any $g \in G$ and all $(a_1, \ldots, a_n) \in \prod_{i=1}^n \mathcal{T}_{k_i}(V, \chi_i)$,

$$g \cdot f_{\ell_1, \ldots, \ell_r}(a_1, \ldots, a_n) = f_{\ell_1, \ldots, \ell_r}(g \cdot a_1, \ldots, g \cdot a_n), \tag{85}$$

and so that the $f_{\ell_1, \ldots, \ell_r}$ are $G$-equivariant. $\qquad\square$

### F.3 REAL EQUIVARIANT TENSOR MAPS (AND PROOF OF THEOREM 2)

All the definitions in the previous subsection can be specialized to the real case. Hence we will have a self-paired real vector space $(V, \langle \cdot, \cdot \rangle)$ on which a group $G$ acts (rationally) in a structure-preserving way. Then we get the family of (rational) $G$-modules:

$$\mathcal{T}_k(V, \chi) := (V^{\otimes k}, \langle \cdot, \cdot \rangle)$$

where $\chi : G \to \mathbb{R}^*$ is a one-dimensional (rational) group-homomorphism of $G$. Together with this family, we have the $k$-contractions given by

$$\iota_k^G : \mathcal{T}_{2k+k'}(V, \chi) \to \mathcal{T}_{k'}(V, \chi), \tag{86}$$

which are $G$-equivariant linear maps. Then we get a very similar theorem to Theorem 4 from which Theorem 2 follows.

**Theorem 5.** Let $G \subset GL(V)$ be either a compact or a complexly averagable real linear algebraic group acting rationally in a structure-preserving way on a self-paired vector space $(V, \langle \cdot, \cdot \rangle)$ and $f : \prod_{i=1}^n \mathcal{T}_{k_i}(V, \chi_i) \to \mathcal{T}_{k'}(V, \chi')$ a $G$-equivariant entire function. Then we may write $f$ as follows:

$$f(a_1, \ldots, a_n) = \sum_{r=0}^{\infty} \sum_{1 \leq \ell_1 \leq \cdots \leq \ell_r \leq n} \iota_{k_{\ell_1, \ldots, \ell_r}}^G \left( a_{\ell_1} \otimes \ldots \otimes a_{\ell_r} \otimes c_{\ell_1, \ldots, \ell_r} \right) \tag{87}$$

where $c_{\ell_1, \ldots, \ell_r} \in \mathcal{T}_{k_{\ell_1, \ldots, \ell_r} + k'}(\mathbb{R}^d, \chi_{\ell_1, \ldots, \ell_r} \chi')$ is a $G$-isotropic tensor for $k_{\ell_1, \ldots, \ell_r} := \sum_{q=1}^r k_{\ell_q}$ and $\chi_{\ell_1, \ldots, \ell_r} = \prod_{q=1}^r \chi_{\ell_q}$.

*Proof of Theorem 2.* This is just a particular case of Theorem 5 as both $O(s, d-s)$ and $Sp(d)$ are both real linear algebraic groups and their complexifications have a Zariski-dense compact subgroup. $\qquad\square$

*Proof of Theorem 5.* When $G$ is compact, we can just repeat the proof for the orthogonal group. When $G$ is a linear algebraic group such that its complexification has a Zariski-dense compact subgroup, we can extend, using the same analytic expression evaluated in the complex tensors, the $G$-equivariant map $f : \prod_{i=1}^n \mathcal{T}_{k_i}(V, \chi_i) \to \mathcal{T}_{k'}(V, \chi')$ to a complex $G^{\mathbb{C}}$-equivariant map $f_{\mathbb{C}} : \prod_{i=1}^n \mathcal{T}_{k_i}(V^{\mathbb{C}}, \chi_i) \to \mathcal{T}_{k'}(V^{\mathbb{C}}, \chi')$. The map becomes $G^{\mathbb{C}}$-equivariant, because $G$ is Zariski-dense inside $G_{\mathbb{C}}$ by Proposition 5.

But for $a \in \prod_{i=1}^n \mathcal{T}_{k_i}(V, \chi_i)$, we have that

$$f(a) = \frac{1}{2} f^{\mathbb{C}}(a) + \frac{1}{2} \overline{f^{\mathbb{C}}(a)}, \tag{88}$$

by reality of the input and output. Hence, by linearity, we can change the non-necessarily real $c_{\ell_1, \ldots, \ell_r}$ by the still $G$-isotropic and real $\frac{1}{2} c_{\ell_1, \ldots, \ell_r} + \frac{1}{2} \overline{c_{\ell_1, \ldots, \ell_r}}$. The latter is $G$-isotropic, finishing the proof. $\qquad\square$

---

[6]The statement is qualitatively different from (Cox et al., 2015, Chapter 1 §1 Proposition 5), but its proof is similar. We only need to use that a univariate entire function which vanishes in an infinite set with an accumulation point has to vanish everywhere.

### F.4 PROOF OF COROLLARY 2

The following proposition is needed to prove the above corollary.

**Proposition 7.** (Roe Goodman, 2009, Theorem 5.3.3) Let $G$ be either $O(s, k - s)$ or $Sp(d)$ and $\langle \cdot, \cdot \rangle$ be the corresponding non-degenerate bilinear form fixed by the usual action of $G$ on $\mathbb{R}^d$, i.e., $\langle \cdot, \cdot \rangle_s$ for $O(s, k - s)$ and $\langle \cdot, \cdot \rangle_{\text{symp}}$ for $Sp(d)$. The subspace of $G$-isotropic tensors in $\mathcal{T}_k(\mathbb{R}^d, \chi_0)$, where $\chi_0$ the constant map to 1, consist only of the zero tensor if $k$ is odd, and it is of the form

$$\sum_{\sigma \in S_k} \alpha_\sigma \left( \theta_G^{\otimes k/2} \right)^\sigma \tag{89}$$

with the $\alpha_\sigma \in \mathbb{R}$ and $\theta_G \in (\mathbb{R}^d)^{\otimes 2}$ the only tensor such that for all $v \in \mathbb{R}^d$, $\iota_1^G(v \otimes \theta_G) = v$, if $k$ is even.

**Remark 9.** Recall that $\theta_G = [\mathbb{I}_{s,d-s}]_{i,j}$ if $G = O(s, d - s)$ and $\theta_G = [J_d]_{i,j}$ if $G = Sp(d)$.

**Remark 10.** Note that the above sum can be written with less summands using the methods of Appendix D.

*Proof of Corollary 2.* By Theorem 2, Proposition 7 and linearity, we can assume, without loss of generality, that

$$f(v_1, \ldots, v_n) = \iota_{r+k} \left( v_{\ell_1} \otimes \cdots \otimes v_{\ell_r} \otimes \theta_G^{\frac{r+k'}{2}} \right)$$

with $1 \leq \ell_1 \leq \cdots \leq \ell_r \leq n$ and $r + k'$ even.

Now, the proof is very similar to that of Corollary 1. However, note that now, we write

$$\theta = e_i \otimes \tilde{e}_i,$$

where $\{e_i \mid i \in [d]\}$ and $\{\tilde{e}_i \mid i \in [d]\}$ are dual basis to each other, i.e., for all $i, j$, $\langle e_i, \tilde{e}_j \rangle = \delta_{i,j}$. The reason we have to pick a couple of bases is that the bilinear form is not necessarily an inner product.

Now, the proof becomes the same as that of Corollary 1, but we have to be careful regarding the $e_i$ and the $\tilde{e}_i$. However, after making the pairings for contraction, we get four cases:

1. $\langle v, e_j \rangle \langle w, \tilde{e}_j \rangle = \pm \langle v, w \rangle$, where the sign depends on whether $\langle \cdot, \cdot \rangle$ is symmetric or antisymmetric.

2. $\langle v, \tilde{e}_j \rangle e_j = v$.

3. $\langle v, e_j \rangle \tilde{e}_j = \pm v$, where the sign depends on whether $\langle \cdot, \cdot \rangle$ is symmetric or antisymmetric.

4. $e_j \otimes \tilde{e}_j = \pm \theta$, where the sign depends on whether $\langle \cdot, \cdot \rangle$ is symmetric or antisymmetric, or $\sum_j \tilde{e}_j \otimes e_j = \theta_G$.

Now, putting these back together as we did in the proof of Corollary 1 gives the desired statement. $\qquad \square$

## G EQUIVARIANCE IN SPARSE VECTOR RECOVERY

In this section, we show a sufficient condition for the $O(d)$-invariance of sparse vector estimation. We start with a lemma on the equivariance of finding an eigenvector.

**Lemma 5.** Let $b$ be a $2_{(+)}$-tensor and let $g \in O(d)$. If $u$ is an eigenvector for eigenvalue $\lambda$ of $M(g) \, b \, M(g)^\top$, then $M(g)^\top u$ is an eigenvector for eigenvalue $\lambda$ of $b$.

*Proof.* Let $b$ be a $2_{(+)}$-tensor, let $g \in O(d)$, and let $\lambda, u$ be an eigenvalue, eigenvector pair of $M(g) \, b \, M(g)^\top$.

$$(M(g) \, b \, M(g)^\top) \, u = \lambda u \Rightarrow b(M(g)^\top u) = \lambda(M(g)^\top u) \, .$$

Thus $M(g)^\top u$ is an eigenvector for eigenvalue $\lambda$ of $b$. $\qquad \square$

**Proposition 8.** Let $S \in \mathbb{R}^{n \times d}$ with rows $a_i^\top \in \mathbb{R}^d$ so that $a_i$ are column vectors. We define the action of $O(d)$ on $S$ for all $g \in O(d)$ as $S M(g)$, and therefore $M(g)^\top a_i$ for the rows. Let $f : \mathbb{R}^{n \times d} \to \mathbb{R}^n, h : (\mathbb{R}^d)^n \to \mathbb{R}^{d \times d}$ symmetric such that $f(S) = S \lambda_{\mathrm{vec}}(h(a_1, \ldots, a_n))$ where $\lambda_{\mathrm{vec}}(\cdot)$ returns a normalized eigenvector for the top eigenvalue of the input symmetric matrix. If $h$ is $O(d)$-equivariant, then $f$ is $O(d)$-invariant.

*Proof.* Let $S, h,$ and $f$ be defined as above. Suppose that $h$ is $O(d)$-equivariant. Suppose $\lambda_{\mathrm{vec}}(M(g)^\top h(a_1, \ldots, a_n) M(g)) = u$, then by lemma 5, up to a sign flip, we have:

$$\lambda_{\mathrm{vec}}(M(g)^\top h(a_1, \ldots, a_n) M(g)) = u = M(g)^\top M(g) u = M(g)^\top \lambda_{\mathrm{vec}}(h(a_1, \ldots, a_n)) \quad (90)$$

Thus,

$$
\begin{aligned}
f(g \cdot S) &= (g \cdot S) \lambda_{\mathrm{vec}}(h(g^{-1} \cdot a_1, \ldots, g^{-1} \cdot a_n)) \\
&= (g \cdot S) \lambda_{\mathrm{vec}}(g^{-1} \cdot h(a_1, \ldots, a_n)) \\
&= S M(g) \lambda_{\mathrm{vec}}(M(g)^\top h(a_1, \ldots, a_n) M(g)) \\
&= S M(g) M(g)^\top \lambda_{\mathrm{vec}}(h(a_1, \ldots, a_n)) \\
&= S \lambda_{\mathrm{vec}}(h(a_1, \ldots, a_n)) \\
&= f(S) .
\end{aligned}
$$

This completes the proof. $\qquad \square$

# H   DERIVATION OF SPARSEVECTORHUNTER (28)

In the following, we derive the general form of an $O(d)$-equivariant function $h : (\mathbb{R}^d)^n \to S_d$ stated in (28) from Corollary 1.

First, we use Corollary 1 to write the arbitrary form of an $O(d)$-equivariant function $g : (\mathbb{R}^d)^n \to \mathbb{R}^{d \times d}$ that takes values in the space of $d \times d$ matrices that are not necessarily symmetric. Given the general form of $g$, it follows that

$$h = \frac{1}{2}(g + g^\top) \quad (91)$$

is the general form of an $O(d)$-equivariant function $h : (\mathbb{R}^d)^n \to S_d$.

In the notation of Corollary 1, we seek an $O(d)$-equivariant function $g : (\mathcal{T}_1(\mathbb{R}^d, +))^n \to \mathcal{T}_2(\mathbb{R}^d, +)$. From Corollary 1 with $k' = 2$, it follows that $g$ can be written in the form

$$g(v_1, \ldots, v_n) = \sum_{t=0}^1 \sum_{\sigma \in S_2} \sum_{1 \leq J_1 \leq \cdots \leq J_{2-2t} \leq n} q_{t, \sigma, J}\left((\langle v_i, v_j \rangle)_{i,j=1}^n\right) \left(v_{J_1} \otimes \cdots \otimes v_{J_{2-2t}} \otimes \delta^{\otimes t}\right)^\sigma . \quad (92)$$

Expanding the sum of the $t = 0$ and $t = 1$ terms, we have

$$g(v_1, \ldots, v_n) = \left( \sum_{\sigma \in S_2} \sum_{1 \leq J_1 \leq J_2 \leq n} q_{0, \sigma, J}\left((\langle v_i, v_j \rangle)_{i,j=1}^n\right) (v_{J_1} \otimes v_{J_2})^\sigma \right) + \sum_{\sigma \in S_2} q_{1, \sigma}\left((\langle v_i, v_j \rangle)_{i,j=1}^n\right) \delta^\sigma. \quad (93)$$

The set of permutation $S_2$ consists of $(1, 2)$ and $(2, 1)$. Using the fact that $(u \otimes v)^{(1,2)} = u \otimes v$, $(u \otimes v)^{(2,1)} = v \otimes u$, and $\delta^\sigma = \delta$ for all $\sigma \in S_2$, we can write the above expression as

$$g(v_1, \ldots, v_n) = \sum_{J_1=1}^n \sum_{J_2=1}^n q_{0, J}\left((\langle v_i, v_j \rangle)_{i,j=1}^n\right) (v_{J_1} \otimes v_{J_2}) + q_1\left((\langle v_i, v_j \rangle)_{i,j=1}^n\right) \delta , \quad (94)$$

where the double sum over $J_1$ and $J_2$ accounts for both the sum over $J_1 \le J_2$ and the sum over the permutations in $S_2$. Next, we swap to standard matrix and vector notation as well as more simple indices to make the equations clearer for readers who are primarily interested in the application. Thus $u \otimes v \Rightarrow uv^\top, \delta \Rightarrow \mathbb{I}_d$ and $J_1, J_2, i, j$ become $i, j, \ell, m$, and we have

$$g(v_1, \ldots, v_n) = \sum_{i=1}^n \sum_{j=1}^n q_{i,j} \left( (\langle a_\ell, a_m \rangle)_{\ell,m=1}^n \right) a_i a_j^\top + q_{\mathbb{I}} \left( (\langle a_\ell, a_m \rangle)_{\ell,m=1}^n \right) \mathbb{I}_d . \quad (95)$$

Finally, setting $h = \frac{1}{2}(g + g^\top)$ gives the desired form of $h$ stated in (28).

## I EXPERIMENTAL DETAILS

### I.1 SYNTHETIC DATA SPARSE VECTOR SAMPLINGS

We consider the following sampling procedures for $v_0$. All these procedures use the same sparsity parameter $\varepsilon \le 1/3$.

**Accept/Reject (A/R).**  A random vector $v_0 \sim \mathcal{N}(\mathbf{0}_n, \mathbb{I}_n)$ is sampled and normalized to unit $\ell_2$ length. We accept it if $\|v_0\|_4^4 \ge \frac{1}{\varepsilon n}$ and otherwise reject it. Note that the sparsity of $v_0$ is not explicitly imposed, but the 4-norm condition suggests that $v_0$ is approximately sparse. The 4-norm condition of sparsity is used in Hopkins et al. (2016).

**Bernoulli-Gaussian (BG)**  This sampling procedure, considered in Mao & Wein (2022), defines $v_0$ as

$$\begin{cases} [v_0]_i = 0 & \text{with probability } 1 - \varepsilon \\ [v_0]_i \sim \mathcal{N}\left(0, \frac{1}{\varepsilon n}\right) & \text{with probability } \varepsilon. \end{cases} \quad (96)$$

Note that under this sampling procedure $\mathbb{E}\|v_0\|_4^4 = \frac{3}{\varepsilon n}$.

**Corrected Bernoulli-Gaussian (CBG)**  We consider a modified version of the Bernoulli-Gaussian that replaces the values set to exactly 0 in the Bernoulli-Gaussian distribution with values sampled from a Gaussian with small variance. Under this distribution we have $\mathbb{E}\|v_0\|_2 = 1$ and $\mathbb{E}\|v_0\|_4^4 = \frac{1}{\varepsilon n}$.

$$\begin{cases} [v_0]_i \sim \mathcal{N}\left(0, \frac{1-\varepsilon-\sqrt{\frac{1}{3}(1-\varepsilon)(1-3\varepsilon)}}{(1-\varepsilon)n}\right) & \text{with probability } 1 - \varepsilon \\ [v_0]_i \sim \mathcal{N}\left(0, \frac{\varepsilon+\sqrt{\frac{1}{3}(1-\varepsilon)(1-3\varepsilon)}}{\varepsilon n}\right) & \text{with probability } \varepsilon. \end{cases}, \quad (97)$$

**Bernoulli-Rademacher (BR)**  This sampling procedure, studied in Mao & Wein (2022), defines $v_0$ as

$$[v_0]_i = \begin{cases} 0 & \text{with probability } 1 - \varepsilon \\ \frac{1}{\sqrt{\varepsilon n}} & \text{with probability } \frac{\varepsilon}{2} \\ \frac{-1}{\sqrt{\varepsilon n}} & \text{with probability } \frac{\varepsilon}{2}. \end{cases} \quad (98)$$

Under this distribution we have $\mathbb{E}\|v_0\|_2 = 1$ and $\mathbb{E}\|v_0\|_4^4 \ge \frac{1}{\varepsilon n}$.

Since the BG, CBG, and BR distributions have $\mathbb{E}\|v_0\|_2 = 1$, we also normalize these vectors to unit $\ell_2$ length after generating them.

**Proposition 9.** Let $v_0$ be a Bernoulli-Gaussian vector. Then $\mathbb{E}\left[\|v_0\|_2^2\right] = 1$ and $\mathbb{E}\left[\|v_0\|_4^4\right] = \frac{3}{\varepsilon n}$.

*Proof.* Let $\varepsilon \in (0, 1]$ and let $v_0$ be a Bernoulli-Gaussian sparse vector. Thus

$$\mathbb{E}\left[\|v_0\|_2^2\right] = \mathbb{E}\left[\sum_{i=1}^n [v_0]_i^2\right] = \sum_{i=1}^n \mathbb{E}\left[[v_0]_i^2\right] . \quad (99)$$

Thus, we need to find the 2nd moment of an entry of $[v_0]_i$, which we will do by first calculating its moment generating function. If $Z$ is a Bernoulli-Gaussian random variable, then $Z = XY$ where $X$

and $Y$ are random variables with $X \sim \text{Bern}(\varepsilon)$ and $Y \sim \mathcal{N}\left(0, \frac{1}{\varepsilon n}\right)$. Then

$$
\begin{aligned}
\mathbb{E}[\exp\{tXY\}] &= \mathbb{E}[\mathbb{E}[\exp\{tXY\}|X]] \\
&= \mathbb{E}[\exp\{tXY\}|X=0]P(X=0) + \mathbb{E}[\exp\{tXY\}|X=1]P(X=1) \\
&= \mathbb{E}[\exp\{0\}]\,(1-\varepsilon) + \varepsilon\mathbb{E}[\exp\{tY\}] \\
&= (1-\varepsilon) + \varepsilon\mathbb{E}[\exp\{tY\}]\,.
\end{aligned}
$$

Since $\mathbb{E}[\exp\{tY\}]$ is the moment generating function of $Y$, a Gaussian random variable, we can see that the 2nd moment of $Z$ is the 2nd moment of $Y$ multiplied by $\varepsilon$. Then

$$
\sum_{i=1}^{n} \mathbb{E}\left[[v_0]_i^2\right] = \sum_{i=1}^{n} \varepsilon\left(\frac{1}{\varepsilon n}\right) = \sum_{i=1}^{n} \frac{1}{n} = 1\,. \tag{100}
$$

Now, for the sparsity condition, we have

$$
\mathbb{E}\left[\|v_0\|_4^4\right] = \sum_{i=1}^{n} \mathbb{E}\left[[v_0]_i^4\right] = \sum_{i=1}^{n} \varepsilon\left(3\left(\frac{1}{\varepsilon n}\right)^2\right) = \sum_{i=1}^{n} \frac{3}{\varepsilon n^2} = \frac{3}{\varepsilon n}\,. \tag{101}
$$

This follows because our previous analysis shows that the 4th moment of an entry of $[v_0]_i$ is $3\sigma^4 = 3\left(\frac{1}{\varepsilon n}\right)^2$. This completes the proof. $\qquad\square$

**Proposition 10.** Let $v_0$ be a Corrected Bernoulli-Gaussian vector. Then $\mathbb{E}\left[\|v_0\|_2^2\right] = 1$ and $\mathbb{E}\left[\|v_0\|_4^4\right] = \frac{1}{\varepsilon n}$.

*Proof.* Let $\varepsilon \in \left(0, \frac{1}{3}\right]$ and let $v_0 \in \mathbb{R}^n$ be a Corrected Bernoulli-Gaussian sparse vector. Thus

$$
\mathbb{E}\left[\|v_0\|_2^2\right] = \mathbb{E}\left[\sum_{i=1}^{n} [v_0]_i^2\right] = \sum_{i=1}^{n} \mathbb{E}\left[[v_0]_i^2\right]\,. \tag{102}
$$

Thus, we need to find the 2nd moment of an entry of $[v_0]_i$, which we will do by first calculating its moment-generating function. If $Z$ is a Corrected Bernoulli-Gaussian random variable, then $Z = XY + (1-X)W$ where $X, Y,$ and $W$ are random variables with $X \sim \text{Bern}(\varepsilon), Y \sim \mathcal{N}\left(0, \frac{\varepsilon+q}{\varepsilon n}\right)$, and $W \sim \mathcal{N}\left(0, \frac{1-\varepsilon-q}{n(1-\varepsilon)}\right)$ where $q = \sqrt{\frac{1}{3}(1-\varepsilon)(1-3\varepsilon)}$. Then

$$
\begin{aligned}
\mathbb{E}[\exp\{t(XY + (1-X)W)\}] &= \mathbb{E}[\mathbb{E}[\exp\{t(XY + (1-X)W)\}|X]] \\
&= \mathbb{E}[\exp\{tW\}|X=0]P(X=0) + \mathbb{E}[\exp\{tY\}|X=1]P(X=1) \\
&= (1-\varepsilon)\mathbb{E}[\exp\{tW\}] + \varepsilon\mathbb{E}[\exp\{tY\}]
\end{aligned}
$$

Since $\mathbb{E}[\exp\{tW\}]$ is the moment generating function of $W$ and $\mathbb{E}[\exp\{tY\}]$ is the moment generating function of $Y$, we can immediately get the moments of $Z$. Then

$$
\begin{aligned}
\sum_{i=1}^{n} \mathbb{E}\left[[v_0]_i^2\right] &= \sum_{i=1}^{n} \varepsilon\left(\frac{\varepsilon+q}{n\varepsilon}\right) + (1-\varepsilon)\left(\frac{1-\varepsilon-q}{n(1-\varepsilon)}\right) \\
&= \sum_{i=1}^{n} \frac{\varepsilon+q+1-\varepsilon-q}{n} \\
&= 1\,.
\end{aligned}
$$

For the sparsity condition, we use the same result above but now for the 4th moment

$$\mathbb{E}\left[\|v_0\|_4^4\right] = \sum_{i=1}^{n} \mathbb{E}\left[[v_0]_i^4\right]$$

$$= \sum_{i=1}^{n} \varepsilon\left(3\left(\frac{\varepsilon+q}{n\varepsilon}\right)^2\right) + (1-\varepsilon)\left(3\left(\frac{1-\varepsilon-q}{n(1-\varepsilon)}\right)^2\right)$$

$$= \sum_{i=1}^{n} 3\varepsilon\left(\frac{\varepsilon^2+2\varepsilon q+q^2}{n^2\varepsilon^2}\right) + 3(1-\varepsilon)\left(\frac{(1-\varepsilon)^2-2(1-\varepsilon)q+q^2}{n^2(1-\varepsilon)^2}\right)$$

$$= \sum_{i=1}^{n} \left(\frac{3\varepsilon^2(1-\varepsilon)+6\varepsilon(1-\varepsilon)q+3(1-\varepsilon)q^2+3\varepsilon(1-\varepsilon)^2-6\varepsilon(1-\varepsilon)q+3\varepsilon q^2}{n^2\varepsilon(1-\varepsilon)}\right)$$

$$= \sum_{i=1}^{n} \left(\frac{3\varepsilon(1-\varepsilon)+3q^2}{n^2\varepsilon(1-\varepsilon)}\right)$$

$$= \sum_{i=1}^{n} \left(\frac{3\varepsilon(1-\varepsilon)+3\left(\frac{1}{3}(1-\varepsilon)(1-3\varepsilon)\right)}{n^2\varepsilon(1-\varepsilon)}\right)$$

$$= \sum_{i=1}^{n} \left(\frac{3\varepsilon+(1-3\varepsilon)}{n^2\varepsilon}\right)$$

$$= \frac{1}{n\varepsilon} \ .$$

This completes the proof. $\qquad\square$

**Proposition 11.** Let $v_0$ be a Bernoulli-Rademacher vector. Then $\mathbb{E}\left[\|v_0\|_2^2\right] = 1$ and $\mathbb{E}\left[\|v_0\|_4^4\right] = \frac{1}{\varepsilon n}$.

*Proof.* Let $\epsilon \in (0,1]$ and let $v_0$ be a Bernoulli-Rademacher sparse vector. Thus

$$\mathbb{E}\left[\|v_0\|_2^2\right] = \mathbb{E}\left[\sum_{i=1}^{n} [v_0]_i^2\right]$$

$$= \sum_{i=1}^{n} \mathbb{E}\left[[v_0]_i^2\right]$$

$$= \sum_{i=1}^{n} (1-\epsilon)(0)^2 + \frac{\epsilon}{2}\left(\frac{1}{\sqrt{\epsilon n}}\right)^2 + \frac{\epsilon}{2}\left(\frac{-1}{\sqrt{\epsilon n}}\right)^2$$

$$= \sum_{i=1}^{n} \frac{\epsilon}{\epsilon n}$$

$$= 1 \ .$$

We also have

$$\mathbb{E}\left[\|v_0\|_4^4\right] = \mathbb{E}\left[\sum_{i=1}^{n} [v_0]_i^4\right]$$

$$= \sum_{i=1}^{n} \mathbb{E}\left[[v_0]_i^4\right]$$

$$= \sum_{i=1}^{n} (1-\epsilon)(0)^4 + \frac{\epsilon}{2}\left(\frac{1}{\sqrt{\epsilon n}}\right)^4 + \frac{\epsilon}{2}\left(\frac{-1}{\sqrt{\epsilon n}}\right)^4$$

$$= \sum_{i=1}^{n} \frac{\epsilon}{\epsilon^2 n^2}$$

$$= \frac{1}{\epsilon n} \ .$$

This completes the proof. □

## I.2 Synthetic Data Noise Vector Covariance

All noise vectors for a single experiment trial are sampled from $\mathcal{N}(0, \Sigma)$ with the same $n \times n$ covariance matrix $\Sigma$. Suppose we want $\Sigma$ that is diagonal but not necessarily the identity matrix. We generate $\Sigma$ by sampling the diagonal entries $[\Sigma]_{i,j} \sim \mathrm{Unif}\left(\frac{1}{2}, \frac{3}{2}\right)$ and setting the off-diagonal entries to 0. Instead, suppose we want $\Sigma$ to be a random covariance matrix. First, we sample an $n \times n$ matrix $M$ with entries $[M]_{i,j} \sim \mathcal{N}(0, 1)$. Then we set $\Sigma = M M^\top + 0.00001 \, \mathbb{I}_n$ which ensures that $\Sigma$ is symmetric and positive definite. In all cases, we don't worry about the scaling because the sampled vectors $v_1, \ldots, v_{d-1}$ will be normalized to unit length.

## I.3 MNIST Noise Schemes

Given a sparse vector $v$ MNIST digit, we construct $v_0, \ldots, v_{d-1}$ using one of the following 4 methods.

**Random Subspace (RND)** Let $v_0 = v$, and sample $v_1, \ldots, v_{d-1} \sim \mathcal{N}(\mathbf{0}_n, \mathbb{I}_n)$, so this method is identical to the synthetic data experiments.

**Gaussian Noise (GAU)** Let $v_i = \frac{v}{\|v\|} + w_i$, where $w_i \sim \mathcal{N}(\mathbf{0}_n, \frac{1}{20}\mathbb{I}_n)$.

**Bernoulli Noise (BER)** Each pixel in each $v_i$ has a 70% chance of taking the value as $v$ and a 30% chance being selected from $\mathrm{Uniform}(0, 255)$, the range of pixel values in the original image.

**Block Noise (BLK)** We select the location of the $12 \times 12$ block uniformly at random and then select the color uniformly from $[0, 255]$. Note that the location of the block is uniformly chosen, so it is more likely that pixels in the middle, where many blocks overlap, will be chosen. The middle of the image is also where the digit pixels are primarily located. One further factor that makes this noise interesting is that the noise itself is sparse in a sense, unlike the other noise schemes.

## I.4 Generating a Random Orthonormal Basis

Now suppose we have a set of vectors $v_0, \ldots, v_{d-1} \in \mathbb{R}^n$, and we would like to get a random orthonormal basis of $\mathrm{span}\{v_0, \ldots, v_{d-1}\}$. To do this, we form the matrix $B$ whose columns are the vectors $v_0, \ldots, v_{d-1}$. If we right multiply $B$ by a random orthogonal matrix and take the Q-R factorization, then $Q$ is a random orthonormal basis of $v_0, \ldots, v_{d-1}$ as we show below. We will add an assumption that the $v_0, \ldots, v_{d-1}$ are linearly independent, which is reasonable given that $d \ll n$ and we are generating these vectors randomly.

**Proposition 12.** Let $n \geq d$, and let $B$ be the $n \times d$ matrix with $v_0, \ldots, v_{d-1}$ as the columns. Assume that $v_0, \ldots, v_{d-1}$ are linearly independent, so rank $B = d$. Let $O$ be a $d \times d$ orthogonal matrix and $Q R = B O$ be a Q-R factorization of $B O$. Then the columns of $Q$ form an orthonormal basis of $\mathrm{span}\{v_0, \ldots, v_{d-1}\}$.

*Proof.* The Q-R factorization gives us that the columns of $Q$ are orthonormal. Thus we just have to show that $\mathrm{span}\{v_0, \ldots, v_{d-1}\} = \mathrm{span}\{Q_0, \ldots, Q_{d-1}\}$

Let $a \in \mathrm{span}\{v_0, \ldots, v_{d-1}\}$, so for some $\alpha_0, \ldots, \alpha_{d-1}$ we have $a = \alpha_0 v_0 + \ldots + \alpha_{d-1} v_{d-1}$. Let $\alpha \in \mathbb{R}^d$ be the vector of these coefficients, and then we have,

$$a = B \alpha = B O O^\top \alpha = Q R O^\top \alpha = Q \hat{\alpha} \,. \tag{103}$$

Thus $\hat{\alpha} \in \mathbb{R}^d$ is a vector of coefficients, so $a \in \mathrm{span}\{Q_0, \ldots, Q_{d-1}\}$. Therefore, $\mathrm{span}\{v_0, \ldots, v_{d-1}\} \subseteq \mathrm{span}\{Q_0, \ldots, Q_{d-1}\}$.

Now let $b \in \mathrm{span}\{Q_0, \ldots, Q_{d-1}\}$, so for some $\beta_0, \ldots, \beta_{d-1}$ we have $b = \beta_0 Q_0 + \ldots + \beta_{d-1} Q_{d-1}$. Let $\beta \in \mathbb{R}^d$ be a vector of the coefficients $\beta_0, \ldots, \beta_{d-1}$. Now, since rank $B = d$, rank $B O = d$, so in the Q-R factorization, the upper triangular $R$ has positive diagonal entries, so it is invertible [Horn & Johnson (1990), Theorem 2.1.14]. Then we have,

$$b = Q \beta = Q R R^{-1} \beta = B O R^{-1} \beta = B \hat{\beta} \,. \tag{104}$$

Thus $\hat{\beta} \in \mathbb{R}^d$ is a vector of coefficients, so $b \in \text{span}\{v_0, \ldots, v_{d-1}\}$. Therefore, $\text{span}\{Q_0, \ldots, Q_{d-1}\} \subseteq \text{span}\{v_0, \ldots, v_{d-1}\}$ which completes the proof.

$\square$

### I.5 Training Details

For synthetic data experiments, the train dataset had $5\,000$ vectors and the validation and test datasets had $500$ vectors. For MNIST, we used a train dataset of $900$ images and validation and test datasets of $100$ images each, all from random classes of digits.

For training our models, we used $1 - \langle \hat{v}, v_0 \rangle^2$ as the loss function. We used the Adam optimizer Kingma & Ba (2017) with exponential decay of $0.999$ per epoch. For the synthetic data experiments, we used a batch size of $100$ and trained until the validation error had not improved for $20$ epochs. For the MNIST experiments, we used a batch size of $10$ and trained for $30$ epochs. See Table 3 for the learning rate and number of parameters for each model. We did a small exploration to find these hyper-parameters. These hyper-parameters seemed to work well, but it is always possible that better ones could be found with more exploration.

| experiment | model | parameter count | learning rate |
|---|---|---|---|
| synthetic | Baseline | $99\,087$ | $1\text{e}{-}3$ |
|  | SVH-Diag | $58\,981$ | $5\text{e}{-}4$ |
|  | SVH | $1\,331\,131$ | $3\text{e}{-}4$ |
| MNIST | Baseline | $2\,067\,282$ | $1\text{e}{-}3$ |
|  | SVH-Diag | $234\,769$ | $5\text{e}{-}4$ |
|  | SVH | $79\,117\,321$ | $3\text{e}{-}4$ |

Table 3: Parameter count and learning rate for each model. Since all models have the same number of hidden layers of the same width, the difference in the number of parameters is driven by the different inputs and outputs of each model.

The experiments were run on a single RTX 6000 Ada GPU and took 18 hours.

### I.6 Further Results

| sampling | $\Sigma$ | SOS-I | SOS-II | BL | SVH-Diag | SVH |
|---|---|---|---|---|---|---|
| A/R | Random | $0.610 \pm 0.011$ | $0.610 \pm 0.011$ | $0.647 \pm 0.177$ | $0.768 \pm 0.045$ | $\mathbf{0.966 \pm 0.001}$ |
|  | Diagonal | $0.444 \pm 0.012$ | $0.444 \pm 0.012$ | $0.561 \pm 0.262$ | $0.698 \pm 0.034$ | $\mathbf{0.755 \pm 0.057}$ |
|  | Identity | $0.611 \pm 0.002$ | $0.611 \pm 0.002$ | $0.494 \pm 0.285$ | $0.622 \pm 0.201$ | $\mathbf{0.647 \pm 0.289}$ |
| BG | Random | $0.963 \pm 0.001$ | $0.963 \pm 0.001$ | $0.783 \pm 0.090$ | $\mathbf{0.970 \pm 0.003}$ | $0.965 \pm 0.002$ |
|  | Diagonal | $0.949 \pm 0.002$ | $0.949 \pm 0.002$ | $0.672 \pm 0.260$ | $\mathbf{0.974 \pm 0.004}$ | $0.775 \pm 0.078$ |
|  | Identity | $0.963 \pm 0.000$ | $0.963 \pm 0.000$ | $0.681 \pm 0.241$ | $0.966 \pm 0.004$ | $\mathbf{0.999 \pm 0.001}$ |
| CBG | Random | $0.409 \pm 0.005$ | $0.409 \pm 0.005$ | $0.836 \pm 0.149$ | $0.490 \pm 0.089$ | $\mathbf{0.965 \pm 0.002}$ |
|  | Diagonal | $0.292 \pm 0.005$ | $0.292 \pm 0.005$ | $\mathbf{0.835 \pm 0.150}$ | $0.597 \pm 0.027$ | $0.722 \pm 0.013$ |
|  | Identity | $0.418 \pm 0.006$ | $0.418 \pm 0.006$ | $0.558 \pm 0.216$ | $0.368 \pm 0.119$ | $\mathbf{0.750 \pm 0.288}$ |
| BR | Random | $0.523 \pm 0.006$ | $0.523 \pm 0.006$ | $\mathbf{0.975 \pm 0.005}$ | $0.669 \pm 0.150$ | $0.970 \pm 0.002$ |
|  | Diagonal | $0.340 \pm 0.010$ | $0.340 \pm 0.010$ | $\mathbf{0.943 \pm 0.008}$ | $0.701 \pm 0.041$ | $0.913 \pm 0.002$ |
|  | Identity | $0.526 \pm 0.005$ | $0.526 \pm 0.005$ | $\mathbf{0.949 \pm 0.006}$ | $0.570 \pm 0.199$ | $0.898 \pm 0.001$ |

Table 4: Synthetic data train error comparison of different methods under different sampling schemes for $v_0$ and different covariances for $v_1, \ldots, v_{d-1}$. The metric is $\langle v_0, \hat{v} \rangle^2$, which ranges from 0 to 1 with values closer to 1, meaning that the vectors are closer. For each row, the best value is **bolded**. For these experiments, $n = 100, d = 5, \epsilon = 0.25$, and the results were averaged over 5 trials with the standard deviation given by $\pm 0.xxx$.

