# OpenReview forum: "Learning equivariant tensor functions with applications to sparse vector recovery"
_ICLR.cc/2025/Conference — ICLR 2025 Conference Withdrawn Submission_

### Official Review · Reviewer_QBDK · 2024-10-28

**Soundness:** 3
**Presentation:** 1
**Contribution:** 2
**Rating:** 5
**Confidence:** 3

**Summary:**

This paper introduces a framework for constructing equivariant tensor functions to facilitate sparse vector recovery in distributions with specific symmetry properties, such as those invariant under orthogonal transformations. The key concepts introduced include the $ k(p)$-tensor space, which captures parity characteristics in tensor components, and the construction of equivariant polynomial functions that remain consistent under orthogonal group actions. The framework also incorporates tensor operations such as outer product and $ k $-contraction to enable flexible tensor combinations while preserving equivariance. Furthermore, the paper extends Haar averaging to non-compact groups, allowing the model to maintain equivariance in settings with broader symmetry requirements, like the indefinite orthogonal group.

**Strengths:**

- To the best of my knowledge, this paper introduces a new approach to equivariant tensor functions, expanding to cases with mixed symmetry properties through the $k(p)$-tensor space concept.

- The paper provides a well-grounded method with precise definitions, such as outer product and $k$-contraction, supported by theoretical proofs.

- Extending Haar averaging to non-compact groups (such as the indefinite orthogonal group $O(s, d-s)$ and symplectic group $Sp(d)$) broadens the versatility.

**Weaknesses:**

1. **Lack of Clear Theoretical Motivation and Practical Relevance**

   - **W1.1:** The paper focuses heavily on mathematical formulas and definitions but lacks intuitive explanations to clarify the need for equivariant tensor functions and their practical motivations.
   - **Suggestion**: Consider adding a high-level overview section that explains why equivariance is essential in certain applications and provides examples that demonstrate the demand for equivariant tensor functions in real-world scenarios.

   - **W1.2:** Key theoretical challenges and the novel aspects of the proposed method are not thoroughly discussed, leaving readers uncertain about the primary obstacles and contributions within the framework.
   - **Suggestion**: Consider adding a dedicated “Contributions” section that explicitly outlines how this method differs from existing approaches, helping readers understand the unique contributions of the framework.

   - **W1.3:** The paper does not clearly explain the necessity of "learning equivariant tensor functions," lacking specific application scenarios that showcase the unique benefits and suitability of these functions. It remains unclear why certain types of tensors in applications require equivariance.
   - **Suggestion**: Include examples of practical applications where specific types of tensors need to be equivariant and clarify how these scenarios validate the advantages of using equivariant tensor functions.

2. **Limitations in Sparse Vector Recovery Application**

   - **W2.1:** The paper centers its applications on sparse vector recovery, a problem that is primarily theoretical and rooted in computer science, with limited overlap with current mainstream machine learning applications, such as computer vision or natural language processing.
   - **Suggestion**: Discuss the potential connections between sparse vector recovery and mainstream machine learning applications, such as compressed sensing, feature selection, or dimensionality reduction in deep learning, to enhance the method’s appeal.

   - **W2.2:** The paper does not demonstrate if or how sparse vector recovery can extend to practical uses in deep learning or other prominent areas within machine learning, lacking case studies or examples of its effectiveness in real-world tasks.
   - **Suggestion**: Consider adding relevant cases or discussions on how the method could be integrated into neural network architectures or applied to tasks such as network pruning, compression, or efficient inference, to illustrate its practicality.

   - **W2.3:** The real-world significance of the proposed method is unclear, potentially making it seem disconnected from practical contributions to machine learning, which could limit its appeal to a broader audience.
   - **Suggestion**: To increase its appeal, the authors could discuss the broader implications of their work, such as how advances in equivariant functions might contribute to more efficient or interpretable machine learning models over the long term.

3. **Limited Relevance to Machine Learning**

   - **W3.1:** While the method shows mathematical novelty, its applications in machine learning are not well-established, which may restrict its potential readership. Machine learning researchers tend to focus on methods with clear, practical applicability and accessibility.
   - **Suggestion**: Strengthen the connection to machine learning demands by discussing how equivariant tensor functions could enhance model interpretability or efficiency, demonstrating its contributions to machine learning research.

   - **W3.2:** The paper extensively uses group theory terms and concepts, such as orthogonal groups and indefinite orthogonal groups, which may be unfamiliar and challenging for machine learning readers without a strong mathematics background.
   - **Suggestion**: Consider including a brief primer on the relevant group theory concepts or offering more intuitive explanations for these mathematical objects. Providing references to introductory resources may also help readers better understand these terms.


**Additional Suggestions:**

- **Add Visual Aids and Case Studies**: Introduce diagrams in appropriate sections to help readers visually understand the basic principles of the method, or add case studies to illustrate its practical applications. This could help engage a wider range of machine learning researchers.

**Questions:**

Please refer to each weakness and suggestion section to help improve the readability and applicability of the work.

---

> ### Author Response · Authors · 2024-11-22
> **Rebuttal to QBDK (Part 1)**
>
> > W1.1: The paper focuses heavily on mathematical formulas and definitions but lacks intuitive explanations to clarify the need for equivariant tensor functions and their practical motivations.
>
> > Suggestion: Consider adding a high-level overview section that explains why equivariance is essential in certain applications and provides examples that demonstrate the demand for equivariant tensor functions in real-world scenarios.
>
> Tensor functions appear in physical applications such as: polarization maps in cosmic microwave backgrounds, stress tensors in material sciences, and potential vorticity (a pseudoscalar or 0(−) -tensor field) in the Earth’s atmosphere (see Figure 1 of [1]). The equivariance appears because all these applications coming from physics have a coordinate-free description, and therefore are equivariant with respect to coordinate transformations.
>
> [1] Gregory, W., Hogg, D. W., Blum-Smith, B., Arias, M. T., Wong, K. W., & Villar, S. (2023). GeometricImageNet: Extending convolutional neural networks to vector and tensor images. arXiv preprint arXiv:2305.12585.
>
> > W1.2: Key theoretical challenges and the novel aspects of the proposed method are not thoroughly discussed, leaving readers uncertain about the primary obstacles and contributions within the framework.
>
> > Suggestion: Consider adding a dedicated “Contributions” section that explicitly outlines how this method differs from existing approaches, helping readers understand the unique contributions of the framework.
>
> See the section entitled “Our contributions” at the end of page 2.
>
> > W1.3: The paper does not clearly explain the necessity of "learning equivariant tensor functions," lacking specific application scenarios that showcase the unique benefits and suitability of these functions. It remains unclear why certain types of tensors in applications require equivariance.
>
> > Suggestion: Include examples of practical applications where specific types of tensors need to be equivariant and clarify how these scenarios validate the advantages of using equivariant tensor functions.
>
> The application of sparse vector recovery is one example where one needs to learn an O(d)-equivariant tensor function. Other examples are the ones described in the answer of your question W1.1.
>
> > W2.1: The paper centers its applications on sparse vector recovery, a problem that is primarily theoretical and rooted in computer science, with limited overlap with current mainstream machine learning applications, such as computer vision or natural language processing.
>
> > Suggestion: Discuss the potential connections between sparse vector recovery and mainstream machine learning applications, such as compressed sensing, feature selection, or dimensionality reduction in deep learning, to enhance the method’s appeal.
> Assignee:
>
> The sparse vector recovery problem has applications to dictionary learning, representation learning, and computer vision tasks. Section VI of [1] gives a thorough discussion of many applications of this problem.
>
> [1] Qu, Q., Zhu, Z., Li, X., Tsakiris, M. C., Wright, J., & Vidal, R. (2020). Finding the sparsest vectors in a subspace: Theory, algorithms, and applications. arXiv preprint arXiv:2001.06970.
>
> > W2.2: The paper does not demonstrate if or how sparse vector recovery can extend to practical uses in deep learning or other prominent areas within machine learning, lacking case studies or examples of its effectiveness in real-world tasks.
>
> > Suggestion: Consider adding relevant cases or discussions on how the method could be integrated into neural network architectures or applied to tasks such as network pruning, compression, or efficient inference, to illustrate its practicality.
>
> This is out of the scope of this paper.
>
> > W2.3: The real-world significance of the proposed method is unclear, potentially making it seem disconnected from practical contributions to machine learning, which could limit its appeal to a broader audience.
>
> > Suggestion: To increase its appeal, the authors could discuss the broader implications of their work, such as how advances in equivariant functions might contribute to more efficient or interpretable machine learning models over the long term.
>
> Equivariant machine learning is a well-established field, with plenty of applications, including AI4physics, AI4science (drug design and protein folding as the most important applications), and graph neural networks. The specific contribution here is theoretical.

---

> > ### Author Response · Authors · 2024-11-22
> > **Rebuttal to QBDK (Part 2)**
> >
> > > W3.1: While the method shows mathematical novelty, its applications in machine learning are not well-established, which may restrict its potential readership. Machine learning researchers tend to focus on methods with clear, practical applicability and accessibility.
> >
> > >Suggestion: Strengthen the connection to machine learning demands by discussing how equivariant tensor functions could enhance model interpretability or efficiency, demonstrating its contributions to machine learning research.
> >
> > Implementing symmetries in machine learning models in the form of equivariances has been shown to increase the accuracy and reduce the sample complexity of machine learning models over non-equivariant baselines. We cite some of these results in our manuscript. See for instance [1,2].
> >
> > [1] Bietti, A., Venturi, L., & Bruna, J. (2021). On the sample complexity of learning under geometric stability. Advances in neural information processing systems, 34, 18673-18684.
> >
> > [2] Behrooz Tahmasebi and Stefanie Jegelka. The exact sample complexity gain from invariances for kernel regression. Advances in Neural Information Processing Systems, 36, 2023.
> >
> > > W3.2: The paper extensively uses group theory terms and concepts, such as orthogonal groups and indefinite orthogonal groups, which may be unfamiliar and challenging for machine learning readers without a strong mathematics background.
> >
> > > Suggestion: Consider including a brief primer on the relevant group theory concepts or offering more intuitive explanations for these mathematical objects. Providing references to introductory resources may also help readers better understand these terms.
> >
> > We refer the reader to textbooks [1] and [2].
> >
> > [1] William Fulton and Joe Harris. Representation theory: a first course, volume 129. Springer Science & Business Media, 2013.
> >
> > [2] Michael M Bronstein, Joan Bruna, Taco Cohen, and Petar Velickovi ˇ c. Geometric deep learning: ´ Grids, groups, graphs, geodesics, and gauges. arXiv preprint arXiv:2104.13478, 2021.

---

> > > ### Comment · Reviewer_QBDK · 2024-11-23
> > >
> > > Many thanks for the authors' response, I would like to maintain my original score.

---

### Official Review · Reviewer_dQUR · 2024-10-29

**Soundness:** 4
**Presentation:** 2
**Contribution:** 4
**Rating:** 8
**Confidence:** 4

**Summary:**

This paper develop a generic framework for defining functions that are equiv-
ariant under the action of classical Lie groups acting diagonally on tensors.
The groups considered include the orthogonal group O(d), the indefinite orthogonal group O(s,k− s), and the symplectic group Sp(d) and other general group actions. The main theoretical contribution is a characterization of
O(d)-equivariant polynomial functions mapping multiple tensor inputs to tensor outputs. The authors prove that any such functions can be expressed as linear
combination of tensor products of the inputs with O(d) isotropic tensors. An
important result is Theorem 1, which provides an explicit parameterization
of O(d)-equivariant polynomial functions. The authors also present a practical
corollary (Corollary 1) for the case where the inputs are vectors and the out-
puts are vector spaces, showing that the equivariant functions can be written
as linear combinations of basis elements formed by permutations of the input
vectors. Furthermore, the author also present a generalization form of general
tensors in Theorem 2 and Corollary 2.

As a proof of concept, They consider the challenge of recovering a planted
sparse vector from a set of vectors forming an orthonormal basis of a sub-
space. By designing a machine learning model that learns an equivariant 2-
tensor (which is the covariance matrix) from data, they use the top eigenvector
of this tensor as an estimator for the sparse vector. The numerical experiments
demonstrate that the learned algorithms outperform state-of-the-art methods.
The models adapt effectively to various noise structures and data sampling
methods.

In physics, Lorentz group O(1,3) is a special case of the equivariant group.
Particularly in general relativity, tensors are used to describe physical quantities such as the curvature of spacetime, energy-momentum distributions. The
authors’ work generally present one of the proof for how to comprehend this
structure in the regime of mathematical tensor products. they also provide a
framework for building physics informed machine learning models that inherently respect Lorentz symmetry and applying for advanced physics studies.

**Strengths:**

## Originality
The paper presents a novel and significant advancement in the field of equiv-
ariant machine learning. Additionally, it uniquely incorporates the indefinite
orthogonal group O(s,k− s) and the symplectic group Sp(d), which are funda-
mental in physics and other scientific domains but have been less explored in the
context of physics informed machine learning. The application of these theo-
retical developments to the sparse vector estimation problem further showcases
originality by demonstrating how algorithms can outperform state-of-the-art
methods in regimes not previously addressed.

## Quality
The paper is of high quality, offering rigorous mathematical formulations and
proofs that underpin the proposed methods. The experiments are well-designed,
and the empirical results convincingly demonstrate the effectiveness.

## Significance
The significance of the work is substantial. By providing explicit parameteriza-
tions for equivariant functions of tensor inputs and outputs, the paper equips
researchers and practitioners with powerful tools. The successful application to
sparse vector es

**Weaknesses:**

## Limited Scope of Experiments
One of the main weaknesses of the paper is the limited scope of the experimental
evaluation. The experiments are primarily focused on the sparse vector estima-
tion problem, which, while important, represents a rather narrow application
domain. The method should have broader applications, such as in physics and
general relativity, but these areas are not sufficiently discussed or explored.

## Clarity
The paper is difficult to follow due to disorganized notation and unclear variable
definitions. The use of coordinates is somewhat messy, which makes the math-
ematical developments hard to track. Variables are often introduced without
proper explanation or context, causing confusion. For example:

• In Definition 4, the indices need more explanation.

• In Theorem 1, how each tensor alk contracts with certain indices (dimen-
sions) of cl1 ,l2 ,...,lr should be explained more clearly.

• In Example 1, there should be more explanation of why only the generic
elements of the G4 group are used while others are contracted.

• In Lemma 1, the meaning and constraints of ασ need clarification.
Moreover, the progression from theoretical concepts to experimental appli-
cations lacks smoothness. The presentation could be improved by organizing
notation more systematically and clearly defining all variables.

## Comparative Analysis
There are other contemporary techniques for sparse vector recovery and tensor
analysis that are not considered. The lack of comparison with a wider array
of methods makes it difficult to fully assess the advantages of the proposed
approach.

**Questions:**

• The statement of Theorem 1 introduces a complex expression involving
linear combinations over isotropic tensors. Could the authors provide more
intuitive explanations or mathematical nature of this theorem?

• The authors state that parameterizing all permutation-invariant polyno-
mial functions may be as challenging as solving the graph isomorphism
problem. Could the authors provide an estimate of the computational
complexity or approximate methods for tackling this problem? can we
push this into high-dimensional settings?

• While MLPs are popular in approximating the polynomial functions, is
there a risk that they might not fully capture the necessary polynomial
structures or equivariance properties?

• The learned SVH models perform better when stringent data assumptions
are not met. Could the authors elaborate on why their models are more
robust in these scenarios? Please provide valuable insights.

• The paper focuses on equivariance. Could you derive some examples for
other groups, such as unitary groups or affine transformations?

---

> ### Author Response · Authors · 2024-11-22
> **Rebuttal to dQUR (Part 1)**
>
> > Limited Scope of Experiments: One of the main weaknesses of the paper is the limited scope of the experimental evaluation. The experiments are primarily focused on the sparse vector estima- tion problem, which, while important, represents a rather narrow application domain. The method should have broader applications, such as in physics and general relativity, but these areas are not sufficiently discussed or explored.
>
> We agree with the reviewer that there are many exciting applications for our model in physics and general relativity, and we look forward to pursuing them in future work. Our main goal in this work was to present a strong theoretical framework along with experiments to show the utility of our methods.
>
> > Clarity: The paper is difficult to follow due to disorganized notation and unclear variable definitions. The use of coordinates is somewhat messy, which makes the math- ematical developments hard to track. Variables are often introduced without proper explanation or context, causing confusion. For example:
>
> > • In Definition 4, the indices need more explanation.
>
> > • In Theorem 1, how each tensor alk contracts with certain indices (dimen- sions) of $c_{l_1, \ldots, l_r}$ should be explained more clearly.
>
> >• In Example 1, there should be more explanation of why only the generic elements of the G4 group are used while others are contracted.
>
> >• In Lemma 1, the meaning and constraints of ασ need clarification. Moreover, the progression from theoretical concepts to experimental appli- cations lacks smoothness. The presentation could be improved by organizing notation more systematically and clearly defining all variables.
>
> Thank you for highlighting these confusing areas, we have added details to clarify the meaning. In particular, we have included a brief example following Definition 4, and we have added an explanation of Theorem 1 to help give an intuitive understanding. Following this example, we have referenced Appendix D justifying why the generic elements of the G4 group suffice. In Lemma 1, the values $\alpha_\sigma$ and $\beta_\sigma$ can take any real value which we clarified in the text. To strengthen the connection between the theory and the experiments, we added the full derivation of our SparseVectorHunter model (Equation (28)) from Corollary 1 to the Appendix.
>
> > The statement of Theorem 1 introduces a complex expression involving linear combinations over isotropic tensors. Could the authors provide more intuitive explanations or mathematical nature of this theorem?
>
> Think of each term Theorem 1 as combining r of the input tensors with the tensor product, then mapping them to the appropriate output tensor order and parity with a linear map. However, since a linear map from a $k(p)$-tensor to a $k'(p')$-tensor can always be written as a tensor product with a $(k+k’)(pp’)$-tensor followed by a $k$-contraction, that is what we do where $c_{l_1,\ldots,l_r}$ is that  $(k+k’)(pp’)$-tensor. Additionally, since the function must be $O(d)$-equivariant, the new tensor that we introduce $c_{l_1,\ldots,l_r}$ must be $O(d)$-isotropic. The sums in the expression then allow us to add up all combinations of $r$ input tensors for $r=0$ to $r=R$. The theorem merely says that this polynomial is enough to construct all tensor equivariant polynomials. We will add an explanation like this to the paper.
>
> > The authors state that parameterizing all permutation-invariant polyno- mial functions may be as challenging as solving the graph isomorphism problem. Could the authors provide an estimate of the computational complexity or approximate methods for tackling this problem? can we push this into high-dimensional settings?
>
> Just to clarify, in Remark 3 we meant that finding all polynomial functions in $n \times n$ matrices that are invariant with respect to simultaneous permutations of rows and columns will distinguish graphs up to isomorphism. If one considers $d$-dimensional point clouds of n points $P \in \mathbb R^{d\times n}$ and asks for polynomials that are simultaneously equivariant with respect to orthogonal transformations O(d) and permutations of the n points, one obtains polynomials on the Gram matrix $P^\top P$ that are invariant with respect to simultaneous permutations of rows and columns. We don’t address this problem in this paper, we say that a possible interesting extension would be to address this problem using the formulation of Corollary 1. A recent paper [1] provides a characterization of such polynomial functions (and more!) using different techniques. It also addresses the high computational complexity of the problem and restricts the machine learning models to low-degree polynomials due to this issue.
>
> [1] Puny, O., Lim, D., Kiani, B., Maron, H., & Lipman, Y. (2023, July). Equivariant polynomials for graph neural networks. In International Conference on Machine Learning (pp. 28191-28222). PMLR.

---

> > ### Author Response · Authors · 2024-11-22
> > **Rebuttal to dQUR (Part 2)**
> >
> > > While MLPs are popular in approximating the polynomial functions, is there a risk that they might not fully capture the necessary polynomial structures or equivariance properties?
> >
> > The formulation of Corollary 1 means that $f$ is equivariant regardless of the nature of the $q_{t,\sigma,J}$ functions because they are scalar functions of only inner products of the vectors $v_i,v_j$, following the results in [1]. Additionally, the basis terms of each sum, which are also necessary for the equivariance, are unaffected by modifying the $q$ functions. Thus the universal approximation property of MLPs [2] means we can safely replace the $q_{t,\sigma,J}$ with MLPs to both approximate the desired polynomial and preserve equivariance.
> >
> > [1] Soledad Villar, David W Hogg, Kate Storey-Fisher, Weichi Yao, and Ben Blum-Smith. Scalars are universal: Equivariant machine learning, structured like classical physics. Advances in Neural Information Processing Systems, 34:28848–28863, 2021.
> >
> > [2] K. Hornik, M. Stinchcombe, H. White, Multilayer feedforward networks are universal approximators, Neural networks 2 (5) (1989) 359–366.
> >
> > > The learned SVH models perform better when stringent data assumptions are not met. Could the authors elaborate on why their models are more robust in these scenarios? Please provide valuable insights.
> >
> > The SVH methods have an innate robustness to different data scenarios because they are learned from the data, while the fixed SOS methods are only proven to be optimal when the data assumptions are met. For the synthetic data experiments, the assumptions are only explicitly met for the accept/reject sparse vector sampling with noise vectors using the identity covariance. If we consider the diagonal, non-identity covariance setting, there is likely an optimal version of the SOS method that gives different weights to each term in the sum based on the specific covariance. Rather than deriving those weights and proving they are optimal, the SVH-Diag simply learns them from the data.
> >
> > > The paper focuses on equivariance. Could you derive some examples for other groups, such as unitary groups or affine transformations?
> >
> > We note that our results apply to the unitary group, which is the intersection of the orthogonal and symplectic groups. However, for affine groups, the techniques in this paper do not apply.

---

> > > ### Comment · Reviewer_dQUR · 2024-11-24
> > >
> > > I thank the authors for their in-depth response, and that cleared up some of my confusions. I still think this work is worthy of publication, so I will maintain my positive score. I hope to see the camera ready version of this paper.

---

### Official Review · Reviewer_BxZQ · 2024-11-04

**Soundness:** 3
**Presentation:** 4
**Contribution:** 2
**Rating:** 5
**Confidence:** 3

**Summary:**

This paper studies polynomial functions from tensor spaces to tensor spaces that are preserved by the action of the Orthogonal group O(d), the indefinite orthogonal group O(s, k-s) or the symplectic group Sp(d). The authors provide a characterization of such polynomial functions in the form of a linear combination of "elementary functions".

As a main application of such result authors propose an algorithm for planted vector recovery, dictionary learning, and experimentally demonstrated that the proposed algorithm outperforms earlier proposed Sum-of-Squares based methods.

Study of such polynomial function is also tangentially motivated by physics applications.

**Strengths:**

The characterization of equivariant polynomial functions studied in this paper is demonstrated to have applications to a planted sparse vector recovery problem which has several practical applications and received a lot of attention over the past decade. In this problem, one is given a linear space $L$ defined by a sparse planted vector $v_1$ and randomly selected complement basis $v_2, ... v_n$. The goal is to recover vector $v_1$ from $L$. The Authors observe that the function $h: L \rightarrow v_1$ is equivariant so can be parametrized using the main characterization proven in this paper and they learn parameters of h by training NN over the training dataset.
I think that this is a new interesting approach to the sparse vector recovery problem.

The proposed characterizations of equivariant polynomial functions may find applications in other areas and may be of its own interest.

**Weaknesses:**

If I understand the proposed approach to the sparse vector recovery problem, for the approach to work one actually need to observe many data samples to be able to train neural networks that define coefficients for the unknown polynomial $h: L \rightarrow v_1$. This may be a strong assumption in many applications and is different from SOS-based methods that can recover $v_1$ from observing only one subspace L and does not require to go through the learning process.

The proof of the main theorem follows standard steps for this sort of problems: problem is easily reduced to homogenous degree r monomials and for homogeneous degree monomials the characterization is obtained by using a somewhat standard group averaging technique. In this sense, the proof of the main result is somewhat "standard" for the literature.

**Questions:**

1. Can you provide a bit more insight into what are the requirements for the training set for your method to be able to train the neural networks defining coefficients in the parametrization of h in Eq 28? Do you need to assume that all training samples have the same planted sparse vector $v_1$? Or is it sufficient to assume that all such vectors $v_1$ are coming from some distribution with nice properties?
2. How many training samples do you anticipate needing to recover $n$-dimensional planted sparse vector?
3. Are you aware of any immediate applications of the provided characterizations for the indefinite orthogonal group O(s, k-s) or the symplectic group Sp(d)?

---

> ### Author Response · Authors · 2024-11-22
> **Rebuttal to BxZQ (Part 1)**
>
> > If I understand the proposed approach to the sparse vector recovery problem, for the approach to work one actually need to observe many data samples to be able to train neural networks that define coefficients for the unknown polynomial $h: L \to v_1$. This may be a strong assumption in many applications and is different from SOS-based methods that can recover $v_1$ from observing only one subspace $L$ and does not require to go through the learning process.
>
> While SOS-based methods only require a single subspace, they make strong distributional assumptions on the input data. That's the reason why it doesn't perform well on the MNIST example, since the MNIST data doesn't satisfy the iid spherical gaussian assumption. By contrast, the machine learning approach allows us to learn a data-driven method that can adapt to a wider variety of settings and perform better. Thus our model is more robust to covariance model misspecification.
>
> > The proof of the main theorem follows standard steps for this sort of problems: problem is easily reduced to homogenous degree $r$ monomials and for homogeneous degree monomials the characterization is obtained by using a somewhat standard group averaging technique. In this sense, the proof of the main result is somewhat "standard" for the literature.
>
> We note that our proofs follow the usual pathways in invariant theory. However, we prove the results because we didn’t find them in the given form anywhere in the literature. Moreover, the results for real-valued invariants are particularly scarce in the literature that focuses mainly on complex-valued invariants.
>
> > Can you provide a bit more insight into what are the requirements for the training set for your method to be able to train the neural networks defining coefficients in the parametrization of $h$ in Eq 28? Do you need to assume that all training samples have the same planted sparse vector ? Or is it sufficient to assume that all such vectors are coming from some distribution with nice properties?
>
> For the synthetic data experiments, we pick a sampling scheme for the sparse vector (accept/reject, Bernoulli-Gaussian, corrected Bernoulli-Gaussian, or Bernoulli-Rademacher), a covariance matrix type (random, diagonal, or identity), then sample the covariance matrix if it is not the identity. Then for each data point, we sample the sparse vector in the manner prescribed and sample the noise vectors from a normal distribution with zero mean and the covariance matrix. So all the planted sparse vectors are different but sampled in the same way, and all the noise vectors are different but have the same covariance. We clarified some of these details in the text.
>
> > How many training samples do you anticipate needing to recover $n$-dimensional planted sparse vector?
>
> While the sample complexity of this problem remains an open question, recent work [1] proves that imposing the symmetries of the problem reduces the dimension of the input space by the dimension of the group. For the sparse vector recovery problem, that would reduce the input dimension from $nd$ to $nd - d(d-1)/2$. For the synthetic data experiments we use 5,000 training points and for MNIST we use 900 training points.
>
> [1] Behrooz Tahmasebi and Stefanie Jegelka. The exact sample complexity gain from invariances for kernel regression. Advances in Neural Information Processing Systems, 36, 2023.

---

> > ### Author Response · Authors · 2024-11-22
> > **Rebuttal to BxZQ (Part 2)**
> >
> > > Are you aware of any immediate applications of the provided characterizations for the indefinite orthogonal group $O(s, k-s)$ or the symplectic group $Sp(d)$?
> >
> > The Lorentz and symplectic groups have applications in physics. For example, recent work [1,2] proposes a Lorentz equivariant transformer to address machine learning questions at the Large Hadron Collider: amplitude regression and jet classification. The authors mention on page 5 of [1] that one of the limitations of the approach is that it cannot express higher-rank tensors:
> >
> > “...this formalism can not represent symmetric rank-2 tensors. For most LHC applications, though, one does not encounter higher-order tensor representations as inputs or outputs, so this is not a substantial limitation. Whether higher-order tensors might be needed for internal representations within a network is an open question.”
> >
> > Our model provides a parameterization for equivariant functions with higher-order (and rank) tensor inputs, outputs, and internal representations. It could potentially be applicable to machine learning problems in high-energy physics.
> >
> > Recent work [3] uses symplectic neural networks to identify phase transitions in equations of state. Higher-order tensors might arise in higher-order differential equations.
> >
> > [1] Brehmer, J., Bresó, V., de Haan, P., Plehn, T., Qu, H., Spinner, J., & Thaler, J. (2024). A Lorentz-Equivariant Transformer for All of the LHC. arXiv preprint arXiv:2411.00446.
> >
> > [2] Spinner, J., Bresó, V., de Haan, P., Plehn, T., Thaler, J., & Brehmer, J. (2024). Lorentz-Equivariant Geometric Algebra Transformers for High-Energy Physics. arXiv preprint arXiv:2405.14806.
> >
> > [3] Kevrekidis, G. A., Serino, D. A., Kaltenborn, A., Gammel, J. T., Burby, J. W., & Klasky, M. L. (2024). Neural Network Representations of Multiphase Equations of State. arXiv preprint arXiv:2406.19957.

---

### Official Review · Reviewer_3Dz3 · 2024-11-05

**Soundness:** 2
**Presentation:** 2
**Contribution:** 2
**Rating:** 5
**Confidence:** 5

**Summary:**

This work characterizes equivariant polynomial functions from tuples of tensor inputs to tensor outputs. The goal behind these characterizations is to define equivariant machine learning models. In particular, they focus on the sparse vector estimation problem. They try to recover a sparse vector from the given an orthonormal basis of subspace spanned by $d$ vectors, where one is the sparse vector to be recovered and the other $d-1$ are just vectors. The recovered sparse vector is stated in (25) and the contributed is how to construct $h$.

**Strengths:**

They construct a equivariant function $h$.

**Weaknesses:**

There are some question you need to solve:

1 Are there some applications about your proposed problem to recover a sparse vector?

2. The representation of $h$ is only related to matrices, but not a complex tensor. Your definitions and analysis are all based on tensors.

3. Can you give a details how to get (28) from Corollary 1?

4. Why you set $h$ to be a equivariant function? What is the advantages?

5. In the numerical experiments about mnist, do you get the similar results if you choose $v_1,\cdots, v_{d-1}$ to be some other vectors which is not so related to the sparse vector?

6. How to get the orthonormal basis for the spanned subspace? Do you get the similar results if you choose different orthonormal basis?

**Questions:**

See weakness.

---

> ### Author Response · Authors · 2024-11-22
>
> > Are there some applications about your proposed problem to recover a sparse vector?
>
> Recovering a sparse vector is an important subtask in many applications in machine learning, such as computer vision, representation learning, and scientific imaging. See section VI of [1] for an overview.
>
> [1] Qing Qu, Zhihui Zhu, Xiao Li, Manolis C. Tsakiris, John Wright, and Ren´e Vidal. Finding the sparsest vectors in a subspace: Theory, algorithms, and applications. arXiv preprint arXiv:2001.06970, 2020.
>
> > The representation of h is only related to matrices, but not a complex tensor. Your definitions and analysis are all based on tensors.
>
> This is correct, the theory we develop in sections 3 and 4 is for general tensor functions of any parity and any order that are equivariant to the orthogonal group, the indefinite orthogonal group, and the symplectic group. We demonstrate the technique with one application where the function has 1-tensor (vector) inputs, a 2-tensor (matrix) output, and is equivariant to the orthogonal group. This is one particular case of our theoretical results, but there are many possible other cases that we plan on exploring in future work.
>
> > Can you give a details how to get (28) from Corollary 1?
>
> Thank you for pointing out this gap in exposition. We have added a full derivation in the Appendix, and referenced it from the main text. In short, we write out Corollary 1 for output order 2 which allows us to break the sum over t and the sum over σ into separate cases and simplify the whole expression. We then swap from tensor notation to standard matrix and vector notation to make the equations clearer for readers who are primarily interested in the application.
>
> > Why you set h to be a equivariant function? What is the advantages?
>
> We set $h$ to be an equivariant function due to the inherent symmetry of the problem, which offers an advantage over non-equivariant ML methods by reducing the sample complexity. Concretely for Problem 1, the input can be any orthonormal basis of span$(v_0, …, v_{d-1})$ for a given sparse vector $v$. In the language of equivariance/invariance, this means that the function that estimates the sparse vector $\hat{v}$ in equation (25) must be invariant to $O(d)$. A sufficient condition for this $O(d)$-invariance is that $h$ is $O(d)$-equivariant (as we show in Appendix G), and we conjecture that this is a necessary condition as well. Thus $h$ is an $O(d)$-equivariant function in the problem, so by learning an explicitly equivariant function, we hope to outperform non-equivariant learned methods. Our hopes are confirmed by our numerical experiments where our equivariant learned models do consistently and often significantly better than non-equivariant learned models.
>
> > In the numerical experiments about mnist, do you get the similar results if you choose $v_1,...,v_{d-1}$ to be some other vectors which is not so related to the sparse vector?
>
> We explore this scenario with the Random Subspace (RND) noise scheme, and we do see similar results. We set the original MNIST digit to be $v_0$, and we sample $v_1,...,v_{d-1}$ from a normal distribution with zero mean and identity covariance. This experimental setup is similar to the synthetic data experiments, and the $v_1,...,v_{d-1}$ are indeed unrelated to the sparse vector. In Table 2 we see that our model SVH (and SVH-Diag) both modestly outperform the SOS methods.
>
> > How to get the orthonormal basis for the spanned subspace? Do you get the similar results if you choose different orthonormal basis?
>
> The orthonormal basis is different every time because it is randomly generated. We construct the $n \times d$ matrix $B$ whose columns are the vectors $v_0,...,v_{d-1}$, then we right multiply by a random orthogonal matrix $O$ before taking the QR decomposition. The matrix $Q$ of the decomposition is an orthonormal basis of the spanned subspace as we prove in Appendix I.4. We have added a pointer to this Appendix to the main text.

---

### Author Response · Authors · 2024-11-22
**Global Response**

We thank the reviewers for their detailed assessment of our work, and their appreciation for the novelty and the impactfulness of our paper. We are encouraged that the reviewers find our theoretical contribution novel, providing “a novel and significant advancement in the field of equivariant machine learning” (Reviewer dQUR) and a “new interesting approach” (Reviewer BxZQ), with “substantial significance” (Reviewer dQUR) and “versatility” (Reviewer QBDK). We are grateful for all the comments and constructive feedback, which will undoubtedly contribute to increasing the overall quality of the paper.

Along with detailed responses to each reviewer, we synthesize below the common questions and responses to all reviewers:
1. Theory: We have clarified some technical statements and provided intuitive explanations of our main results, addressing questions from reviewers 3Dz3, dQUR, and QBDK. Each term of Theorem 1 can be rightly seen as a linear map of the tensor product of $r$ chosen input tensors. Corollary 1 simplifies this situation so that the inputs are only vectors, and the equation is a linear combination where the basis elements are tensor products of the vectors and Kronecker deltas.
2. Experiments: Our main goal is to present a theoretical framework to define equivariant machine learning models mapping from tensors to tensors. Our experiments are intended to translate our theory into practical recipes, illustrating the utility of our proposed equivariant machine learning approach compared to classical sum-of-squares methods and non-equivariant ML methods. We agree with the reviewers that our framework can be applied to many other applications such as physics and representation learning, which we intend to pursue in future work.
3. Exposition: We answered several questions about the experimental training set up and added references for where to find the additional details in the Appendix. We also added a new section in the Appendix with the full derivation of our SparseVectorHunter model from Corollary 1 in response to questions from reviewers 3Dz3 and dQUR.

We have made updates to the paper, and uploaded the new version. All additions are written in red.

---

### Author Response · Authors · 2024-12-04

Thank you to all reviewers for helping improve our work. For the reviewers who haven’t had a chance yet to read through our rebuttals, we hope we were able to address any lingering questions and concerns. We are encouraged that the reviewers appreciate the strong theoretical contributions to the field of equivariant machine learning. We believe that this work also provides a concrete recipe for practitioners to use our methods and achieve the same success for equivariant problems. We are excited by future applications in physics and other fields.

---

### Note · Authors · 2025-01-23

I have read and agree with the venue's withdrawal policy on behalf of myself and my co-authors.